# LncRNA EPR controls epithelial proliferation by coordinating *Cdkn1a* transcription and mRNA decay response to TGF-β

Martina Rossi[1,2], Gabriele Bucci [3], Dario Rizzotto[4], Domenico Bordo[1], Matteo J. Marzi[5], Margherita Puppo[1,2], Arielle Flinois[6], Domenica Spadaro[6], Sandra Citi [6], Laura Emionite[7], Michele Cilli[7], Francesco Nicassio [5], Alberto Inga[4], Paola Briata[1] & Roberto Gherzi [1]

Long noncoding RNAs (lncRNAs) are emerging as regulators of fundamental biological processes. Here we report on the characterization of an intergenic lncRNA expressed in epithelial tissues which we termed EPR (Epithelial cell Program Regulator). EPR is rapidly downregulated by TGF-β and its sustained expression largely reshapes the transcriptome, favors the acquisition of epithelial traits, and reduces cell proliferation in cultured mammary gland cells as well as in an animal model of orthotopic transplantation. EPR generates a small peptide that localizes at epithelial cell junctions but the RNA molecule per se accounts for the vast majority of EPR-induced gene expression changes. Mechanistically, EPR interacts with chromatin and regulates *Cdkn1a* gene expression by affecting both its transcription and mRNA decay through its association with SMAD3 and the mRNA decay-promoting factor KHSRP, respectively. We propose that EPR enables epithelial cells to control proliferation by modulating waves of gene expression in response to TGF-β.

[1] Gene Expression Regulation Laboratory, IRCCS Ospedale Policlinico San Martino, 16132 Genova, Italy. [2] DIMES Sezione Biochimica-Università di Genova, 16132 Genova, Italy. [3] Center of Translational Genomics and Bioinformatics, IRCCS Ospedale San Raffaele, 20132 Milano, Italy. [4] Laboratory of Transcriptional Networks, Center for Integrative Biology, CIBIO, University of Trento, 38123 Trento, Italy. [5] Center for Genomic Science of IIT@SEMM, Istituto Italiano di Tecnologia (IIT), 20139 Milano, Italy. [6] Department of Cell Biology, University of Geneve, 1211 Geneve, Switzerland. [7] Animal Facility, IRCCS Policlinico San Martino, 16132 Genova, Italy. These authors jointly supervised this work: Paola Briata, Roberto Gherzi. Correspondence and requests for materials should be addressed to A.I. (email: alberto.inga@unitn.it) or to P.B. (email: paola.briata@hsanmartino.it) or to R.G. (email: rgherzi@ucsd.edu)

Human transcriptome analysis has revealed the existence of a surprisingly high number of noncoding RNAs that have been classified in multiple families based on their size and biogenesis. Long noncoding RNAs (lncRNAs) are defined as transcripts longer than 200 nucleotides transcribed by RNA polymerase II and commonly originated from intergenic regions. LncRNAs can be capped, spliced, and polyadenylated and usually show limited protein coding potential (refs. [1,2], and literature cited therein).

LncRNAs are emerging as a fundamental aspect of biology due to their ability to reprogram gene expression and influence distinct cellular functions including cell fate determination, cell cycle progression, apoptosis, and aging[1,2]. Their expression is usually tissue restricted, developmentally regulated, and can change under specific pathological conditions. Many lncRNAs influence hallmarks of cancer such as uncontrolled proliferation, evasion of cell death, as well as metastasis formation and it has been suggested that lncRNAs can act as oncogenes or tumor suppressors —either directly or indirectly— by interfering with different pathways[3,4]. From a mechanistic point of view, lncRNAs may influence the function of transcriptional complexes, modulate chromatin structures, serve as scaffolds to form ribonucleoprotein (RNP) complexes or as decoys for proteins and micro-RNAs (miRNAs)[2,5]. Thus, lncRNA-mediated control of gene expression may take place at transcriptional and/or posttranscriptional levels[5–9].

Recently, lncRNAs have been described as important components of the transforming growth factor β (TGF-β) signaling pathway[10,11]. TGF-β belongs to a large family of structurally related cytokines that regulate growth, survival, differentiation, and migration of many cell types including mammary gland epithelial cells (ref. [12]. for a review). TGF-β activates membrane kinase receptors and induces phosphorylation of cell-specific SMAD proteins that, in complex with the common SMAD4, accumulate into the nucleus to regulate gene expression at different levels (ref. [13]. for a recent review).

In our previous studies, we showed that the multifunctional RNA-binding protein KHSRP acts as a regulatory hub that conveys extracellular stimuli into gene expression changes due to its ability to interact with several molecular partners[14]. KHSRP is able to posttranscriptionally regulate gene expression by promoting decay of unstable mRNAs, favoring maturation of select miRNAs from precursors, and controlling alternative splicing events[14]. Recently, we reported that KHSRP affects the alternative splicing of a cohort of pre-mRNAs that encode regulators of cell adhesion and motility—such as CD44 and FGFR2—favoring their epithelial type exon usage and that miRNA-mediated KHSRP silencing is required for TGF-β-induced epithelial-to-mesenchymal transition (EMT) in immortalized NMuMG mammary gland cells[15]. Further, we found that Resveratrol—a natural polyphenolic compound endowed with anti-inflammatory, antiproliferative, as well as proapoptotic activities—prevents TGF-β-dependent KHSRP downregulation. thus shifting Cd44 and Fgfr2 pre-mRNA alternative splicing from the mesenchymal-specific to the epithelial-specific isoforms[16]. Our previous observation that the lncRNA H19 interacts with KHSRP and affects its mRNA decay-promoting function[17] prompted us to identify additional KHSRP/lncRNAs interactions endowed with regulatory potential.

Here we describe a previously uncharacterized mammalian lncRNA expressed in epithelial tissues that we termed EPR (after Epithelial Program Regulator). EPR came to our attention due to its ability to interact with KHSRP and to counteract TGF-β-induced EMT. EPR contains an open reading frame (ORF) that is translated into a small peptide localized at epithelial cell junctions. However, we found that EPR regulates the expression of a large set of target transcripts independently of the peptide

biogenesis. Our studies have revealed that EPR interacts with chromatin, regulates Cdkn1a gene expression by affecting both its transcription and mRNA decay, and controls cell proliferation in both immortalized and transformed mammary gland cells as well as in a mouse model of orthotopic transplantation.

## Results

**Identification of EPR, an epithelial cell-enriched lncRNA.** This study was initiated in an attempt to identify lncRNAs which are able to interact with KHSRP and whose expression is regulated by TGF-β in immortalized murine mammary gland NMuMG cells. To this end, we leveraged RNA-sequencing (RNA-Seq) and anti-KHSRP RNP complexes Immunoprecipitation followed by RNA-sequencing (RIP-Seq) analyses performed in untreated or TGF-β-treated NMuMG cells. TGF-β treatment significantly reduced or increased the levels of 110 and 194 lncRNAs, respectively (|log2 fold changes| > 2.0, $p < 0.01$ (Student's $t$ test); Supplementary Table 1a) while RIP-Seq analysis showed that TGF-β modulates the interaction of KHSRP with 67 lncRNAs (|log2 fold changes| > 2.0, $p < 0.01$ (Student's $t$ test); Supplementary Table 1b). Among a set of lncRNA candidates of potential interest in EMT, we focused on the previously uncharacterized BC030870 (ENSMUSG00000074300, located on mouse chromosome 8 and transcribed in reverse orientation) that we renamed EPR (highlighted in yellow in Supplementary Table 1a and 1b). RIP analysis followed by quantitative RT- PCR (qRT-PCR) as well as band-shift analysis confirmed that EPR directly interacts with KHSRP (Supplementary Fig. 1a, b). TGF-β induced a small increase in EPR levels followed by rapid downregulation (Fig. 1a) that accounts for the reduced interaction between KHSRP and EPR upon a 6-h treatment (Supplementary Table 1b). TGF-β-dependent modulation of EPR expression requires TGF-β type I receptor signaling as shown by the ability of SB431542 (a selective inhibitor of ALK5, 4, and 7 [18]) to abrogate the effect of the cytokine on EPR expression (Supplementary Fig. 1c). SMAD complexes are major effectors of TGF-β-dependent transcriptional regulation[13] and our ChIP-qPCR showed that SMAD3 interacts with EPR promoter in a TGF-β-modulated way (Supplementary Fig. 1d, upper panel). Positive (Serpine1) and negative (Mettl9) controls for ChIP experiments are provided in Supplementary Fig. 1d (lower panel) and Supplementary Fig. 1e, respectively. Our data are consistent with the hypothesis that SMAD3 interacts with a corepressor complex on EPR promoter region to modulate its transcription[19]. De novo protein synthesis is not required for TGF-β-induced downregulation of EPR expression as revealed by the use of cycloheximide (Supplementary Fig. 1f, upper panel; Zeb2 (also known as SIP1) represents the control for cycloheximide activity[20]).

EPR is expressed during embryonic development (Supplementary Fig. 1g) and in epithelial tissues of adult mice with a prevalence in the gastrointestinal tract, lung, kidney and mammary gland (Fig. 1b). EPR is polyadenylated and spliced (Supplementary Fig. 1h) and it is almost equally distributed in the cytoplasm, nucleoplasm, and chromatin of NMuMG cells (Fig. 1c; see Supplementary Fig. 2a for an immunoblot-based validation of cell fractionation). LINC01207 (a.k.a. SMIM31, located on chromosome 4 and transcribed in forward orientation; hereafter indicated as h.EPR) is the human ortholog of EPR and displays superimposable epithelial tissue-enriched expression (as evaluated through the Human BodyMap 2.0 data from Illumina; Supplementary Fig. 2b). Bioinformatics analysis performed on RNA-Seq data derived from different subpopulations of normal breast cells isolated by FACS analysis from reduction mammoplasty specimens[21] revealed that h.EPR is expressed exclusively in differentiated luminal cells of the mammary gland (Fig. 1d).

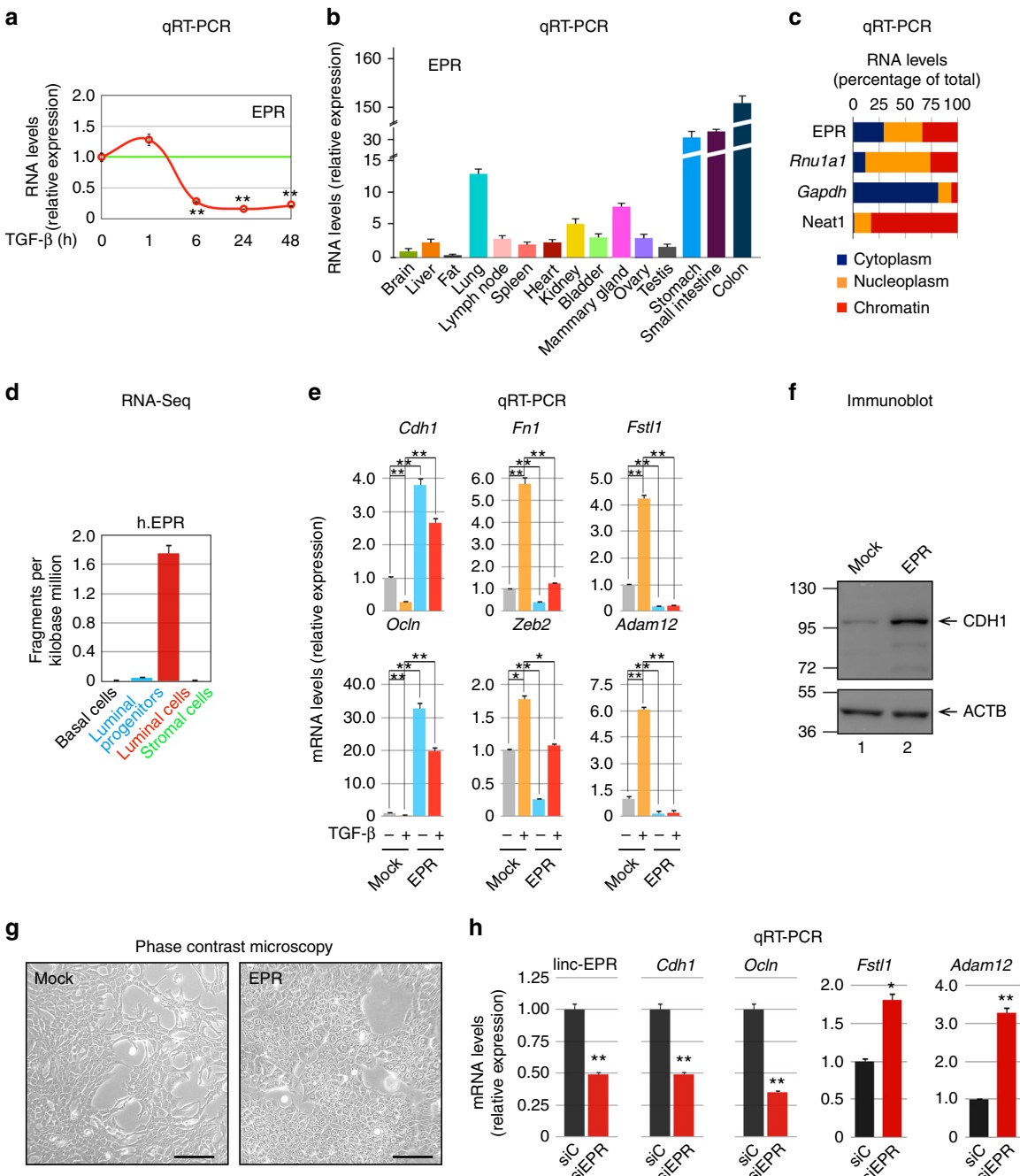

**Fig. 1** EPR displays epithelial expression and antagonizes TGF-β-induced EMT in mammary gland cells. **a** Quantitative RT-PCR (qRT-PCR) analysis of EPR in NMuMG cells serum-starved (2% FBS, 16 h) and either treated with TGF-β (10 ng ml⁻¹) for the indicated times or untreated (time 0). **b** qRT-PCR analysis of EPR in the indicated mouse tissues. **c** NMuMG cells were fractionated and RNA was prepared from cytoplasm, nucleoplasm, and chromatin and analyzed by qRT-PCR to quantify the indicated RNAs. *Rnu1a1* is also known as U1 small nuclear RNA, *Gapdh* mRNA encodes the glyceraldehyde-3-phosphate dehydrogenase. **d** qRT-PCR analysis of h.EPR in normal human breast cells isolated from reduction mammoplasty specimens[21]. **e** qRT-PCR analysis of the indicated transcripts in either mock or EPR-overexpressing (EPR) NMuMG cells serum-starved and either treated with TGF-β (+) for 24 h or untreated (−). **f** Immunoblot analysis of total cell extracts from either mock or EPR-overexpressing (EPR) NMuMG cells. The indicated antibodies were used. The position of molecular mass markers is indicated on the left. Representative gels are shown. ACTB is also known as Actin Beta. **g** Phase contrast microscopy of either mock or EPR-overexpressing (EPR) NMuMG cells. Scale bars: 100 μm. **h** qRT-PCR analysis of the indicated transcripts in NMuMG cells transiently transfected with either control siRNA (siC) or siRNA designed to silence EPR expression (siEPR). The values of qRT-PCR experiments shown are averages (±SEM) of three independent experiments performed in triplicate. Statistical significance: *$p < 0.01$, **$p < 0.001$ (Student's $t$ test)

In order to investigate the potential role of EPR in TGF-β-induced EMT, we decided to counteract TGF-β-dependent EPR downregulation by stably overexpressing the lncRNA in NMuMG cells (overexpression was 3- to 12-fold compared to the respective mock cells (empty vector-transfected), in different transfectant

pools). EPR overexpression prevented TGF-β-induced downregulation of epithelial factors (*Cdh1*, *Ocln*) and induction of mesenchymal markers (*Fn1*, *Fstl1*, *Zeb2*, *Adam12*) as well morphological changes (Fig. 1e, Supplementary Fig. 2c). Strikingly, we observed that EPR overexpression affects the levels of

epithelial and mesenchymal markers (Fig. 1f, Supplementary Fig. 2d), and induced a cobblestone-like cell morphology in untreated cells (Fig. 1g). Further, EPR overexpression significantly limited the migratory potential of NMuMG cells (Supplementary Fig. 2e). Conversely, transient silencing of EPR downregulated the mRNA levels of epithelial markers, enhanced the levels of mesenchymal markers (Fig. 1h), and rescued the gene expression changes induced by stable overexpression of the lncRNA (Supplementary Fig. 2f). Interestingly, bioinformatics analysis of RNA-Seq data derived from human normal breast samples revealed a statistically significant positive correlation between the expression of h.EPR and epithelial markers such as CDH1 and OCLN and a negative correlation with mesenchymal markers such as VIM and SNAI1 (Supplementary Fig. 2g). This observation is in agreement with the evidence that EPR expression is mutually exclusive with the expression of the EMT factor *Cdh2* as revealed by bioinformatics analysis of datasets derived from single-cell RNA-Seq analysis performed in mice (Supplementary Fig. 2h).

In conclusion, the name EPR that we assigned to lncRNA BC030870 (after Epithelial Program Regulator) is consistent with its enriched expression in epithelial cells and with the upregulation of epithelial markers and downregulation of mesenchymal markers induced by its overexpression.

**EPR encodes a small polypeptide**. A few recent reports show that certain lncRNAs contain short ORFs that can be translated into peptides endowed with regulatory functions[22–25]. The analysis of EPR sequence revealed the presence of a 213 nucleotide-long ORF potentially encoding a 71-amino acid polypeptide that, interestingly, corresponds to the lncRNA region that displays the highest identity with the human ortholog (Fig. 2a). The putative polypeptide sequence is well conserved among mammalian species and in silico methods identified a conserved α-helical transmembrane domain while a predicted second α-helix was found in the putative cytosolic domain (Fig. 2b). Importantly, polysome fractionation followed by qRT-PCR analysis revealed that EPR localizes to actively translating polysomes (Supplementary Fig. 3a).

To investigate whether EPR ORF is translated, we inserted a FLAG tag at its 3′ end and transiently transfected the resulting construct into HEK-293 cells (Fig. 2c, left). As shown in Fig. 2c (right), the ORF was translated into a short polypeptide of the expected molecular mass. To unambiguously prove the existence of the endogenous small EPR-encoded peptide (EPRp), the ORF was expressed in bacteria and the resulting peptide was purified and utilized as immunogen to generate a rabbit polyclonal antibody. Polyclonal anti-EPRp recognized a recombinant polypeptide transiently expressed in HEK-293 cells (Supplementary Fig. 3b) and, most importantly, a ~8 KDa polypeptide in mouse gastrointestinal tract organs and breast (Supplementary Fig. 3c). In keeping with EPR downregulation upon TGF-β treatment, the expression of the EPRp was downregulated in response to treatment with TGF-β for 24 h (Fig. 2d).

In order to identify the molecular partners of EPRp, we performed immunoaffinity purification of proteins interacting with EPRp in NMuMG cells. Mass spectrometry (MS) analysis of coimmunoprecipitating proteins separated by SDS-PAGE (Fig. 2e) revealed an enrichment in junctional and cytoskeletal proteins (Supplementary Data 1). Coimmunoprecipitation experiments confirmed that EPRp interacts with the tight junction proteins TJP1 (ZO-1) and CGN (Cingulin), with the tight and adherens junction protein CGNL1 (Paracingulin) as well as with the actin-associated proteins CTTN (Cortactin) and MYH9 (epithelial myosin-II) (Fig. 2f, Supplementary Fig. 3d).

To investigate EPRp subcellular localization, we performed immunofluorescence experiments in NMuMG cells stably transfected with either EPRp-FLAG or with a construct in which the second codon of the ORF—encoding glutamic acid, E, of EPRp—was mutagenized in order to obtain a STOP codon (see also below, EPRSTOPE-FLAG). Specific localization of FLAG signal at cell−cell junctions, labeled by the junctional marker CGN, was detected in cells stably expressing EPRp-FLAG (arrows in Fig. 2g) while no junctional FLAG labeling was detected in mock-transfected cells or in cells expressing the point-mutant version unable to produce the peptide. CGN labeling was wavy and discontinuous in mock-transfected cells and in cells expressing EPRSTOPE-FLAG, whereas it was linear and uninterrupted in cells expressing EPRp-FLAG, suggesting that EPRp overexpression promotes epithelial junction assembly and reorganization of the junction-associated actin cytoskeleton. A weak diffuse cytoplasmic staining observed in NMuMG cells expressing EPRp-FLAG might reflect EPRp interaction with cytoskeletal proteins (Fig. 2g).

On the basis of these results, we conclude that an ORF present in EPR is translated into a small peptide that is well conserved among species and that displays a junctional localization in mammary gland cells.

**EPR regulates gene expression in NMuMG cells**. We set out to investigate the function(s) of EPR in NMuMG cells. First, in order to answer the question whether the phenotypic changes that we observed by overexpressing EPR were caused by the lncRNA per se, the peptide or both, we performed transcriptome-wide RNA-Seq analyses in mock cells as well as in NMuMG cells overexpressing either EPR or a point-mutant version unable to produce the peptide (EPRSTOPE, for details see above and Fig. 3a). Bioinformatics analyses of RNA-Seq data revealed a vast rearrangement of the transcriptome as a consequence of both EPR and EPRSTOPE overexpression (Supplementary Data 2). Gene ontology (GO) analysis of RNA-Seq results revealed the enrichment of terms related to epithelial morphogenesis, cell motility, cell migration, and epithelial cell proliferation among the top regulated categories. Representative examples of transcripts either upregulated or downregulated by both EPR and EPR-STOPE are shown in Fig. 3b, c. In keeping with the sequence conservation between EPR and h.EPR, the overexpression of the human lncRNA in murine NMuMG cells yielded gene expression changes superimposable to those obtained by overexpressing the murine lncRNA (Supplementary Fig. 4a).

Interestingly, overexpression of either EPR or EPRSTOPE caused largely overlapping gene expression changes when compared to mock cells (Fig. 3d, upper panel). When we directly compared gene expression changes induced by EPR or EPR-STOPE by applying stringent statistical criteria, we noticed that only a relatively small group of genes displayed expression changes dependent on the presence of EPRp (Fig. 3d, lower panel). The analysis of three independent NMuMG transfectant pools overexpressing EPRSTOPE, followed by qRT-PCR-based validation, allowed us to further restrict the number of transcripts whose levels are affected by the peptide per se (Fig. 3e, Supplementary Fig. 4b). These include transcripts encoding a calcium-dependent cell adhesion protein (*Pcdh19*), two ion transporters (*Slc9a2*, *Scl39a4*), a cytokine receptor (*Fgfr2*) as well as a modulator of membrane transport and actin dynamics (*Anxa6*). Further, analysis of an additional EPR mutant (referred to as EPRSTOPM in which the start codon has been mutagenized to a STOP codon, see below) confirmed the restricted number of gene expression changes that can be ascribed to the peptide translation (Supplementary Fig. 4c).

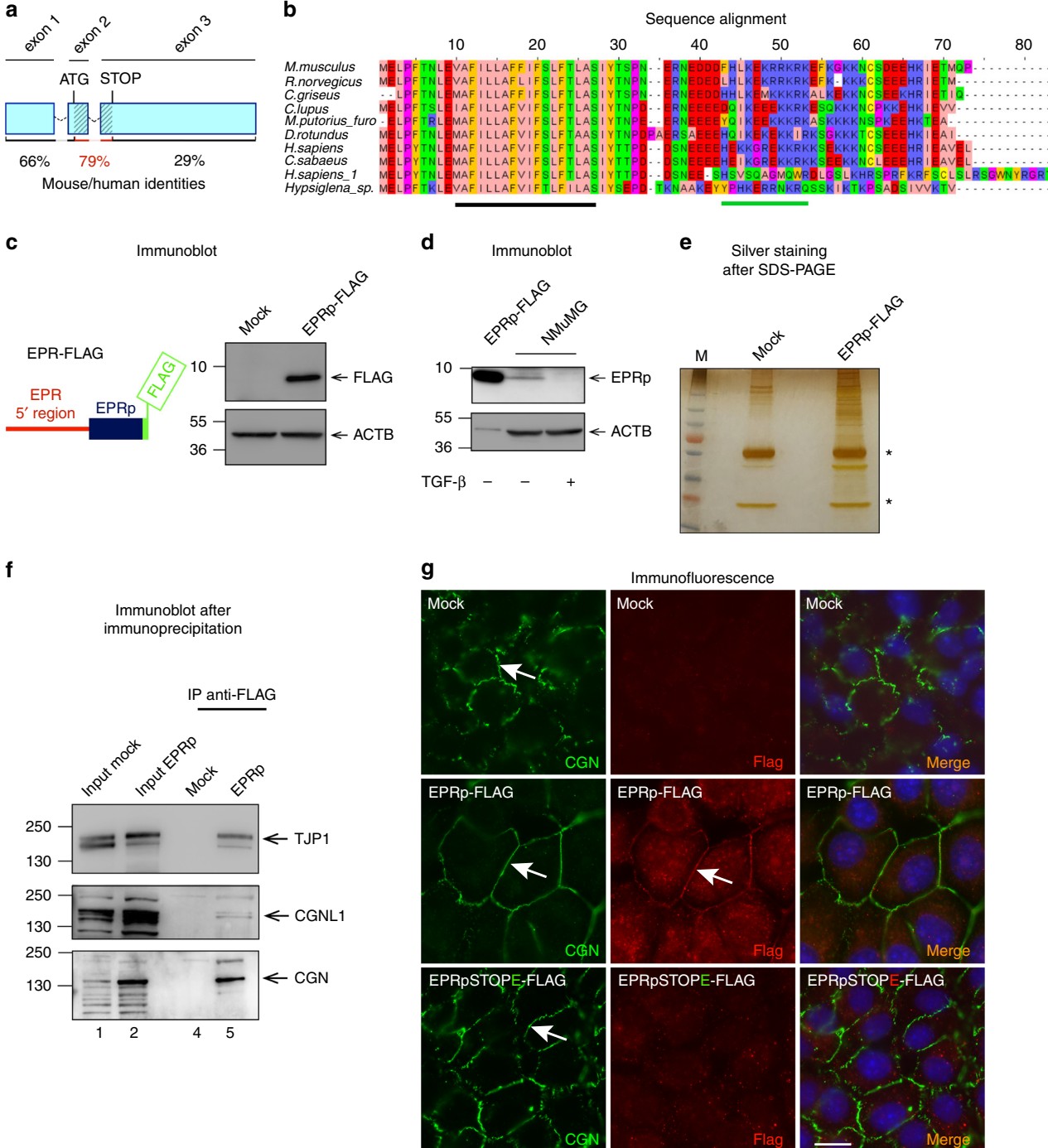

**Fig. 2** A small peptide (EPRp) originates from EPR and displays junctional localization. **a** Schematic of the exon−intron structure (not in scale) of EPR, a putative ORF is dashed. Percentage of human/mouse identity in the putative ORF and in its flanking RNA regions is presented. **b** Alignment of the predicted mammalian amino acids sequence encoded by the putative ORF present in EPR and in its orthologs. The position of the predicted transmembrane α-helix is shown as a solid black bar while the position of an additional cytoplasmic α-helix is shown as a solid green bar. **c** Left, diagram of the FLAG-fusion construct used for transfection (EPRp-FLAG); right, immunoblot analysis of total cell extracts from either mock or EPRp-FLAG-transfected HEK-293 cells. **d** Immunoblot analysis of total cell extracts from NMuMG cells serum-starved and either treated with TGF-β (+) for 24 h or untreated (−); extracts from EPRp-FLAG-transfected HEK-293 cells (EPRp-FLAG) represent a positive control. **e** SDS-PAGE analysis of total cell extracts from either mock or EPRp-FLAG-overexpressing (EPRp-FLAG) NMuMG cells immuno-purified using anti-FLAG monoclonal antibody. A representative silver-stained gel is shown. Asterisks indicate the position of immunoglobulin heavy and light chains. **f** Coimmunoprecipitation of FLAG-tagged EPRp and distinct junctional proteins (as indicated) in total extracts from NMuMG cells stably transfected with EPRp-FLAG. **g** Immunofluorescence analysis of either mock or EPRp-FLAG- or EPRpSTOPE-FLAG-stably transfected NMuMG cells cultured to confluence, to allow formation of cell−cell junctions. Arrows point to CGN and EPRp-FLAG junctional localization. Scale bar: 10 μm. For immunoblots, the indicated antibodies were used; the position of molecular mass markers is presented on the left and representative gels are shown. ACTB is also known as Actin Beta

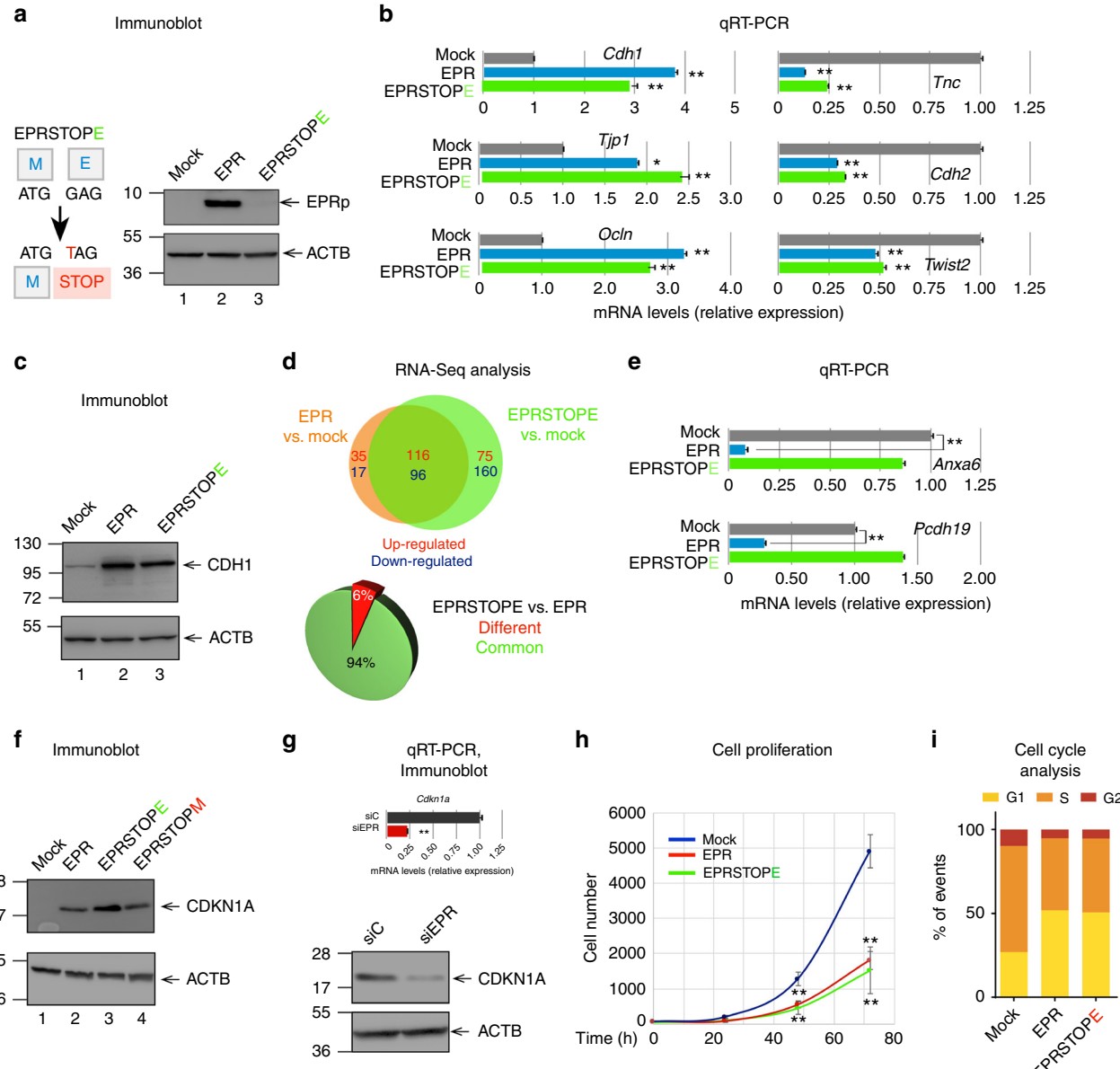

**Fig. 3** EPR overexpression reshapes NMuMG cells transcriptome and reduces cell proliferation. **a** Schematic of the point mutation introduced in the second codon of the EPR ORF to generate EPRSTOPE (left panel). HEK-293 cells were transiently transfected with either EPR or EPRSTOPE and the translation of EPR was assessed by immunoblotting (right panel). **b** qRT-PCR analysis of the indicated transcripts in either mock, EPR- or EPRSTOPE-expressing NMuMG cells. **c** Immunoblot analysis of total cell extracts from either mock, EPR- or EPRSTOPE-overexpressing NMuMG cells. **d** Upper panel, TopTable Venn diagram of the limma one-factor contrast analysis showing the overlap between transcripts either upregulated (numbers in red) or downregulated (numbers in blue) in cells stably overexpressing either EPR or EPRSTOPE compared with mock cells. Only transcripts displaying |log2 fold expression difference| > 2.5 ($p < 0.0001$, Student's $t$ test) were included in further comparisons. Lower panel, Pie diagram showing the percentage of gene expression changes common to NMuMG cells overexpressing EPR and EPRSTOPE as well as the percentage of changes that can be specifically attributed to the EPR translation. **e** qRT-PCR analysis of the indicated transcripts in either mock, EPR- or EPRSTOPE-overexpressing NMuMG cells. **f** Immunoblot analysis of total cell extracts from either mock, EPR-, EPRSTOPE- or EPRSTOPM-overexpressing NMuMG cells. **g** qRT-PCR analysis of *Cdkn1a* mRNA levels in either control siRNA (siC) or siEPR-transfected NMuMG cells (upper panel). Immunoblot analysis of total cell extracts from either control siRNA (siC) or siEPR-transfected NMuMG cells (lower panel). **h** Proliferation analysis (Operetta CLS High-Content Analysis System) of either mock, EPR- or EPRSTOPE-overexpressing NMuMG cells. **i** Cell cycle distribution in cultures grown at similar density (near 90% confluence). Cell cycle was analyzed by flow cytometry after double staining with EdU and propidium iodide of either mock, EPR- or EPRSTOPE-overexpressing NMuMG cells. Bars plot the relative proportion of cells in G1, S, and G2 phases for each cell line. The values of qRT-PCR experiments shown are averages (±SEM) of three independent experiments performed in triplicate. Statistical significance: *$p < 0.01$, **$p < 0.001$ (Student's $t$ test). For immunoblots, the indicated antibodies were used; the position of molecular mass markers is indicated on the left and representative gels are shown

Considering the emerging evidence that some lncRNAs act locally (in *cis*) to regulate the expression of nearby genes, we investigated this possibility and RNA-Seq analysis revealed that the expression levels of genes proximal to EPR (*Palld*, *Cpe*, *Sc4mol*, *Klhl2*, *Tmem192*, *Tma16*, *Naf16*, *Nat2* and *Pssd*, localized over 8 MB of chromosome 8) are unaffected by the almost complete EPR downregulation that occurs in NMuMG cells treated with TGF-β for 24 h (R.G. and G.B., unpublished observation).

Altogether, transcriptome-wide analyses showed a EPR-dependent wide rearrangement of the transcriptome in NMuMG cells with relatively restricted effects on gene expression ascribed to the peptide. Thus, we decided to focus our further studies on the EPR functions that are independent of the peptide biogenesis.

**EPR regulates *Cdkn1a* gene expression and cell proliferation**. Among the GO terms significantly enriched by the overexpression of either EPR or EPRSTOPE, we identified the category Regulation of Epithelial Cell Proliferation. Indeed, both EPR and EPRSTOPE overexpression significantly affected the levels of a group of transcripts belonging to this category including the cyclin-dependent kinase inhibitor *Cdkn1a* (a.k.a. p21$^{WAF1/Cip1}$) (Supplementary Fig. 4d). Immunoblots presented in Fig. 3f show that overexpression of either EPR or EPRSTOPE or EPRSTOPM strongly enhanced CDKN1A levels. Conversely, EPR silencing strongly reduced CDKN1A expression (Fig. 3g). As expected, CDKN1A levels were enhanced by overexpression of the human ortholog of EPR (Supplementary Fig. 4e). Most important, we found that overexpression of either EPR or EPRSTOPE as well as of h.EPR strongly reduces cell proliferation rate in NMuMG cells (Fig. 3h, Supplementary Fig. 4f). Cell cycle analysis demonstrated a relevant increment of cells arrested in the G1 phase in the case of NMuMG cells transfected with either EPR or EPRSTOPE in comparison to mock cells (Fig. 3i). To exclude the possibility that gene expression changes that we observed (Fig. 3b) might be dependent on the EPR-induced G1 arrest, we sorted cells in the G1 phase and analyzed gene expression changes by qRT-PCR. Data presented in Supplementary Fig. 4g indicate that the expression changes induced in G1-enriched cells by overexpression of either EPR or EPRSTOPE are superimposable to those observed in the total cell population (Fig. 3b).

Together, these results provide evidence that modulation of EPR levels regulates *Cdkn1a* gene expression and affects cell proliferation in NMuMG cells. Given the role of CDKN1A in promoting cell cycle arrest in response to many stimuli—including TGF-β[26]—we decided to focus our further mechanistic studies on the role of EPR in TGF-β-dependent regulation of *Cdkn1a* gene expression.

**EPR regulates TGF-β-dependent *Cdkn1a* gene expression**. Analysis of newly synthesized transcripts revealed that overexpression of either EPR or EPRSTOPE strongly enhances *Cdkn1a* transcription (Fig. 4a) and the kinetic analysis of mRNA decay indicated that overexpression of either EPR or EPRSTOPE induces also a significant stabilization of *Cdkn1a* mRNA (Fig. 4b).

TGF-β signaling promotes tissue growth and morphogenesis during embryonic development while, as tissues mature, many cell types gain the ability to respond to TGF-β with growth arrest that is primarily due to imbalance of G1 events[27]. As similarly reported in other cell types[28,29], treatment of NMuMG cells with TGF-β for 1 h caused a rapid induction of *Cdkn1a* gene expression that was followed by return to baseline levels after 6 h (Fig. 4c). The observation that *Cdkn1a* return to baseline levels matches EPR downregulation (Fig. 4c) and that EPR overexpression strongly enhances *Cdkn1a* levels, prompted us to hypothesize a role for EPR in the TGF-β-dependent modulation of *Cdkn1a* gene expression. Our ChIP-qPCR assays showed that TGF-β treatment for 1 h stimulates the binding of SMAD3 to *Cdkn1a* promoter that returns to basal levels after 6 h (Fig. 4d, see also ref. [28]). TGF-β-dependent control of *Cdkn1a* mRNA decay was never investigated in detail but, considering that cells often achieve rapid changes of gene expression by integrating gene transcription control with regulated mRNA decay[30,31], we addressed the possibility that TGF-β could affect *Cdkn1a* mRNA decay. Figure 4e showed that *Cdkn1a* mRNA stability is unaffected by 1 h of TGF-β treatment (upper panel) but is reduced when the treatment is prolonged up to 6 h (lower panel). Thus, the TGF-β-dependent rapid fluctuations of *Cdkn1a* expression depend on the regulation of both transcription and mRNA decay in NMuMG cells.

Our hypothesis that EPR plays a role in the regulation of TGF-β-dependent CDKN1A expression was supported by the evidence that EPR silencing abrogated *Cdkn1a* mRNA induction upon TGF-β treatment for 1 h (Fig. 4f) while its overexpression enhances *Cdkn1a* levels and blunts its rapid modulation by TGF-β (Fig. 4g).

Together, our results indicate that EPR plays a dual role in TGF-β-dependent *Cdkn1a* gene expression control.

**EPR affects both *Cdkn1a* gene transcription and mRNA decay**. We investigated the molecular mechanism(s) by which EPR regulates *Cdkn1a* gene transcription. The evidence of enhanced RNA-Pol II occupancy and reduced presence of the H3K27me3 repressive mark at the *Cdkn1a* promoter in EPR-overexpressing cells (Supplementary Fig. 5a) together with our finding that EPR is present in the chromatin fraction (see Fig. 1c) prompted us to explore the possibility that EPR affects *Cdkn1a* transcription through direct interaction with its promoter region. Chromatin Isolation by RNA Purification (ChIRP)-Seq experiments (P.B., G.B., E. Zapparoli et al., unpublished) as well as ChIRP-qPCR experiments revealed the direct interaction of EPR with *Cdkn1a* promoter (Fig. 5a). The interaction of EPR with *Cdkn1a* promoter is not significantly affected by a 1 h TGF-β treatment (Supplementary Fig. 5b).

RIP experiments showed that SMAD3 interacts with EPR and the interaction is enhanced by treatment with TGF-β for 1 h (Supplementary Fig. 5c). In keeping with growing evidence suggesting that the interaction between lncRNAs and specific transcription factors can affect gene expression[5,32], ChIP-qPCR experiments showed that EPR overexpression enhances SMAD3-*Cdkn1a* promoter association and abrogates its dismissal after 6 h of TGF-β treatment (Fig. 5b). These effects are reproduced by overexpression of EPRSTOPE (Fig. 5b). Cell treatment with SB431542 abrogated the TGF-β-dependent enhancement of SMAD3-*Cdkn1a* promoter association in mock as well as in NMuMG cells overexpressing either EPR or EPRSTOPE (Supplementary Fig. 5d). Notably, EPR overexpression favored SMAD3-*Cdkn1a* promoter interaction also in untreated cells and this was not modified by SB431542 treatment (Fig. 5b and Supplementary Fig. 5d). To explain the association of SMAD3 with *Cdkn1a* promoter in cells overexpressing EPR also in the absence of TGF-β treatment, we hypothesize that EPR overexpression favors the association of SMAD3 molecules present in NMuMG cell nuclei of untreated cells with *Cdkn1a* promoter. Indeed, it is known that, although SMAD proteins rapidly accumulates into nuclei upon TGF-β treatment[33], a certain amount of SMAD3 is present in the nuclei of untreated cells (ref. [34]; Supplementary Fig. 5e).

In keeping with results shown in Fig. 4b, we found that EPR overexpression prevents *Cdkn1a* mRNA destabilization induced by a treatment with TGF-β for 6 h (Fig. 5c). Our initial observation that EPR interacts with KHSRP, a factor able to

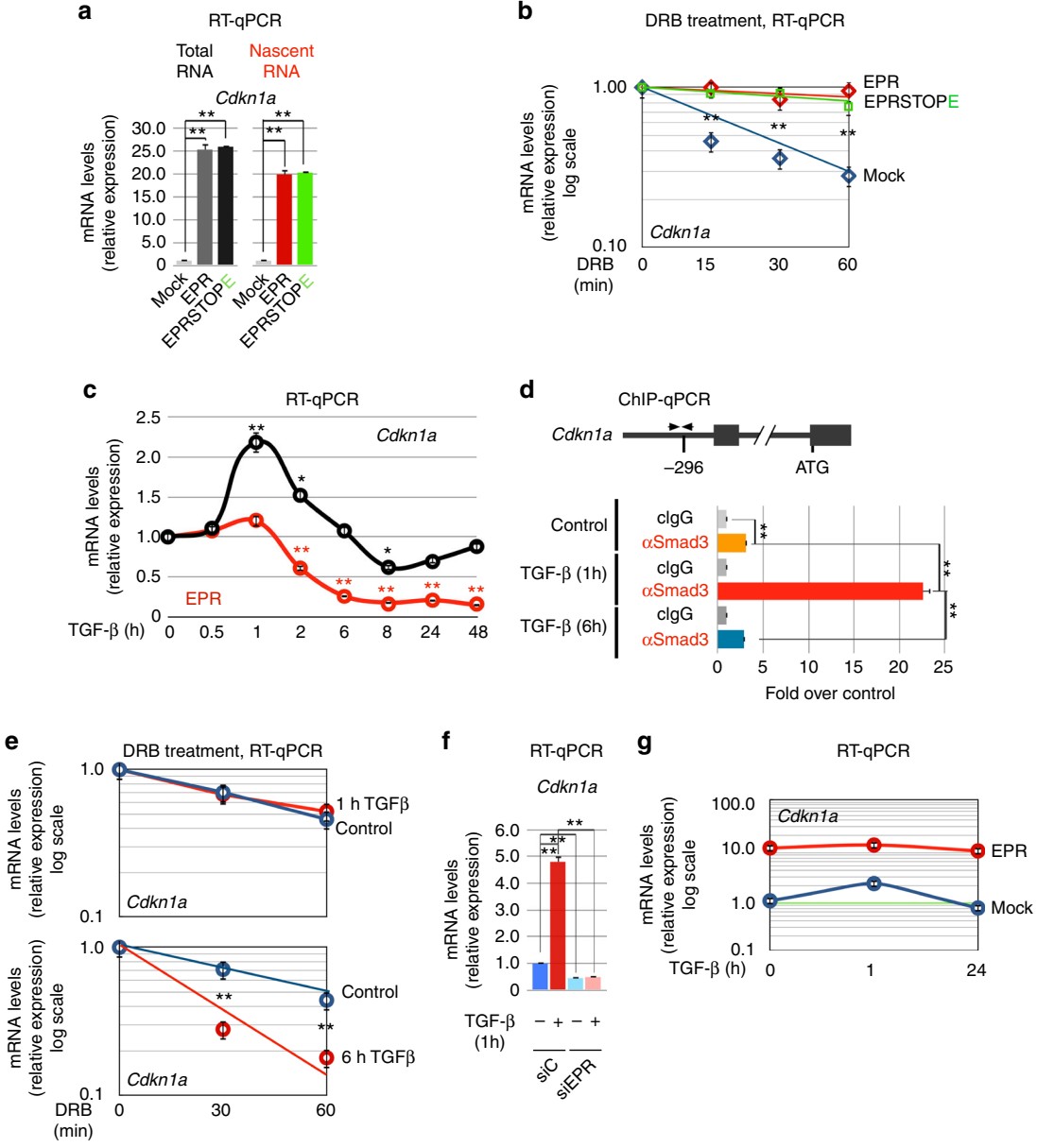

**Fig. 4** Dual regulation of *Cdkn1a* gene expression by EPR. **a** qRT-PCR analysis of either total (left) or nascent (right) *Cdkn1a* transcript in either mock, EPR- or EPRSTOPE- overexpressing NMuMG cells. **b** Either mock, EPR- or EPRSTOPE-overexpressing NMuMG cells were treated with 100 μM 5,6-dichlorobenzimidazole 1-β-ᴅ-ribofuranoside (DRB) for different times (as indicated). *Cdkn1a* gene expression was analyzed by qRT-PCR. **c** qRT-PCR analysis of *Cdkn1a* (black line) and EPR (red line) in NMuMG cells serum-starved (2% FBS, 16 h) and either treated with TGF-β (10 ng ml⁻¹) for the indicated times or untreated (time 0). **d** Chromatin prepared from NMuMG cells serum-starved and either treated with TGF-β for the indicated times or untreated (control) was immunoprecipitated using either control IgG or affinity-purified anti-SMAD3 rabbit polyclonal antibody. The association of SMAD3 with *Cdkn1a* promoter (schematic on the top) was quantified by qPCR (primers indicated as arrowheads in the schematic above). **e** NMuMG cells were serum-starved (2% FBS, 16 h) and either treated with TGF-β (10 ng ml⁻¹) for either 1 h (top panel) or 6 h (bottom panel) or left untreated (control in both panels). Subsequently, cells were treated with 100 μM DRB for the indicated times and total RNA was isolated and analyzed by qRT-PCR to quantify *Cdkn1a* mRNA levels. Please note the slight, reproducible difference in the decay kinetic of *Cdkn1a* mRNA between nontransfected and mock-transfected NMuMG cells. **f** qRT-PCR analysis of *Cdkn1a* in NMuMG cells transiently transfected with either control siRNA (siC) or siRNA designed to silence EPR expression (siEPR), serum-starved (2% FBS, 16 h) and then either treated with TGF-β (10 ng ml⁻¹) for 1 h (+) or left untreated (−). **g** qRT-PCR analysis of *Cdkn1a* expression in either mock (blue line) or EPR overexpressing (red line) NMuMG cells serum-starved (2% FBS, 16 h) and either treated with TGF-β (10 ng ml⁻¹) for the indicated times or left untreated (time 0). Please note the logarithmic scale of the *Y*-axis. The values of both qRT-PCR and qPCR experiments shown are averages (±SEM) of three independent experiments performed in triplicate. Statistical significance: *$p < 0.01$, **$p < 0.001$ (Student's *t* test)

promote rapid decay of select labile mRNAs in many cell types[35], prompted us to explore whether KHSRP regulates *Cdkn1a* mRNA decay in NMuMG cells. KHSRP silencing induced *Cdkn1a* mRNA accumulation and prevented its rapid degradation (Supplementary Fig. 6a and Fig. 5d, upper panel) while

transient KHSRP overexpression in NMuMG cells stably expressing EPR promoted *Cdkn1a* mRNA destabilization (Fig. 5d, lower panel). KHSRP is predominantly nuclear in NMuMG cells[15] and we found that mature *Cdkn1a* mRNA is abundant in nuclear fractions of these cells (Supplementary Fig. 6b) where it

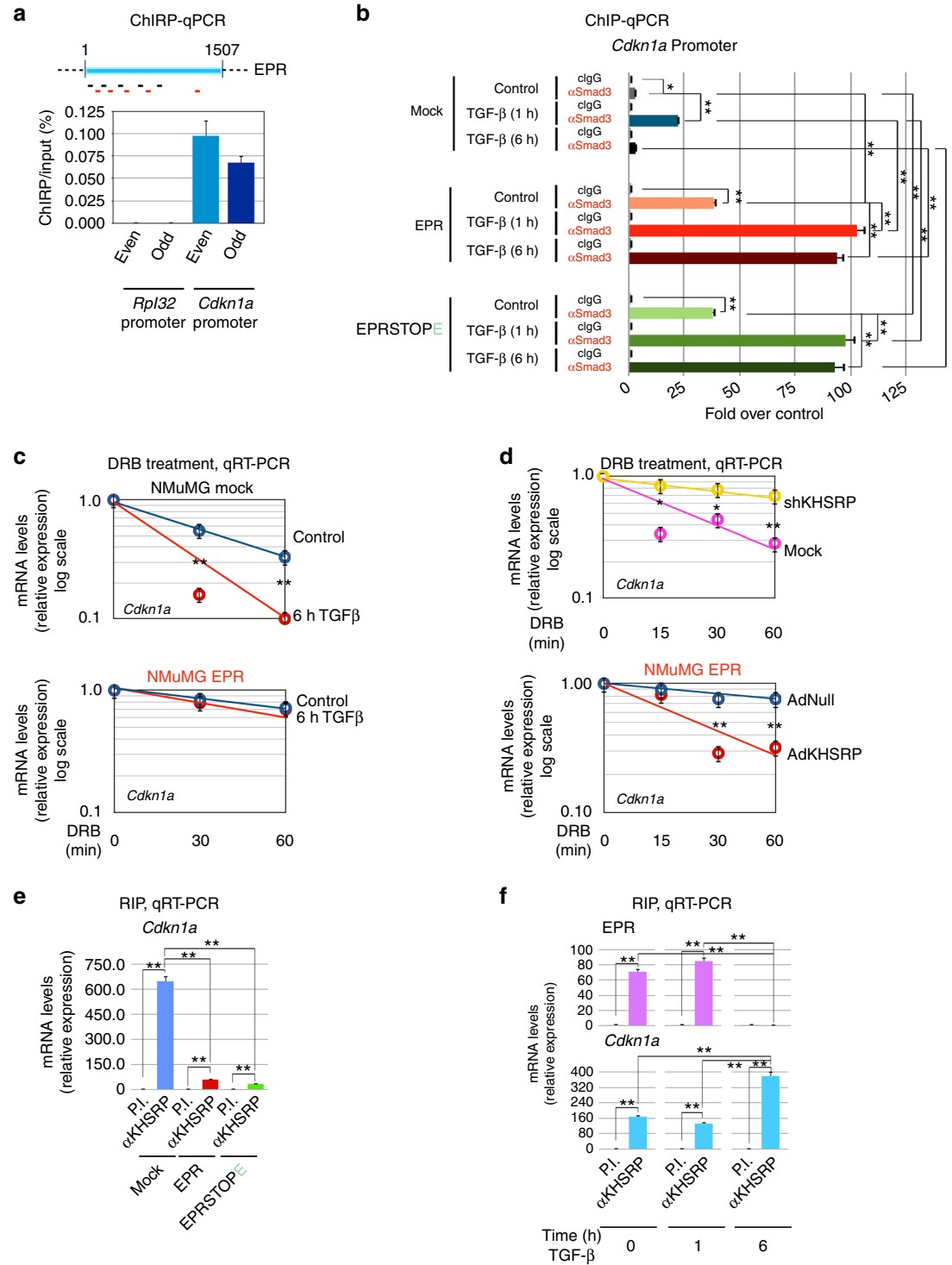

undergoes rapid decay and is stabilized by EPR overexpression as measured by two independent techniques (Supplementary Fig. 6c, d). These findings suggested that EPR might interfere with the ability of KHSRP to interact with *Cdkn1a* mRNA. RIP experiments presented in Fig. 5e show that KHSRP association with *Cdkn1a* mRNA was strongly reduced by EPR and EPRSTOPE overexpression. Based on these results, we investigated whether EPR downregulation, that is physiologically obtained by TGF-β-treatment, affects the interaction of KHSRP with *Cdkn1a* mRNA. RIP analyses indicated that

KHSRP-EPR association is abrogated while KHSRP interaction with *Cdkn1a* mRNA is increased upon TGF-β treatment (6 h) (Fig. 5f).

Altogether, these results suggest that EPR controls *Cdkn1a* expression by sustaining its transcription and by impairing its mRNA decay in response to TGF-β.

**EPR overexpression reduces breast cancer cell proliferation.** We investigated EPR in murine breast cancer cell lines and

**Fig. 5** EPR associates with *Cdkn1a* promoter affecting its transcription as well as its mRNA decay. **a** ChIRP analyses. Top panel is a schematic of EPR and shows the location of biotinylated odd (black) and even (red) tiling oligonucleotides used for ChIRP. Both input and purified DNA were analyzed by qPCR to amplify either *Rpl32* (negative control) or *Cdkn1a* promoters. Values are averages (±SEM) of three independent experiments performed in triplicate. **b** Chromatin was prepared from either mock, EPR- or EPRSTOPE-overexpressing NMuMG cells serum-starved and either treated with TGF-β (1 ng ml$^{-1}$) for the indicated times or left untreated (control). Chromatin was immunoprecipitated using either control IgG or affinity-purified anti-SMAD3 rabbit polyclonal antibody. The association of SMAD3 with *Cdkn1a* promoter was verified by qPCR. **c** Either mock (top panel) or EPR-overexpressing (bottom panel) NMuMG cells were serum-starved, either treated with TGF-β (10 ng ml$^{-1}$) for 6 h or left untreated (control) and then treated with 100 μM DRB and total RNA was isolated and analyzed by qRT-PCR to quantify *Cdkn1a* mRNA levels. **d** Top panel, either mock or shKHSRP NMuMG cells were treated with 100 μM DRB. Total RNA was isolated at different times (as indicated) and analyzed by qRT-PCR to quantify *Cdkn1a* mRNA levels. Bottom panel, NMuMG cells stably overexpressing EPR were infected with either control (AdNull) or KHSRP-expressing (AdKHSRP) adenoviral vectors for 24 h then treated with 100 μM DRB. Total RNA was isolated at different times (as indicated) and analyzed by qRT-PCR to quantify *Cdkn1a* mRNA levels. NMuMG mock cells used for the experiment depicted in the upper panel differ from those presented throughout this report and have been previously described[15]. **e** Total extracts from either mock, EPR- or EPRSTOPE-overexpressing NMuMG cells were immunoprecipitated as indicated. RNA was purified from immunocomplexes and analyzed by qRT-PCR to quantify *Cdkn1a* mRNA levels. **f** Total extracts were prepared from NMuMG cells serum-starved and either treated with TGF-β (10 ng ml$^{-1}$) or left untreated (time 0) and immunoprecipitated as indicated. RNA was purified from immunocomplexes and analyzed by qRT-PCR. The values of both qRT-PCR and qPCR experiments shown are averages (±SEM) of three independent experiments performed in triplicate. Statistical significance: *$p < 0.01$, **$p < 0.001$ (Student's *t* test)

observed that its expression is severely reduced when compared with immortalized NMuMG cells (Supplementary Fig. 7a). Similarly, the expression of h.EPR was below detection levels in highly aggressive human breast cancer cell lines (M.J.M. and F.N., unpublished observation). H.EPR could be detected in about 75% of breast cancer primary samples (780/1043 cases from The Cancer Genome Atlas (TCGA) database[36]; Supplementary Fig. 7b) and, according to PAM50 molecular subtype classification, it was more expressed in Luminal A and Her2 tumors while it was almost absent in Basal-like tumors, the most frequent subtype of triple-negative breast cancers[37] (Fig. 6a).

Based on these observations, we decided to express EPR in triple-negative mesenchymal-like breast cancer cells, such as murine 4T1 and human MDA-MB-231 cell lines, respectively (Supplementary Fig. 7c).

Overexpression of either EPR or EPRSTOPE in 4T1 cells resulted in a strong induction of *Cdkn1a* gene expression as well as in a significant reduction of clonogenic potential, cell proliferation, and anchorage-independent cell growth (Fig. 6b-e). EPR overexpression in 4T1 cells also downregulated the expression of mesenchymal factors such as *Cdh2* and *Adam12* (Supplementary Fig. 7d). Very similar results were observed by overexpressing either human or murine EPR in human MDA-MB-231 cells (Fig. 6f, g, Supplementary Fig. 7e−g).

Finally, to interrogate the activity of EPR on cell proliferation control in vivo, we orthotopically injected either mock or EPR-expressing 4T1 cells into syngenic BALB/c mice. In concordance with our observations in cultured cells, EPR expression resulted in a remarkable reduction of tumor volume after 10 days (Fig. 6h, left panel). A significant reduction of the tumor mass was still evident and statistically significant also at 2 weeks after the transplant when mice were sacrificed (Fig. 6h, right panel, Supplementary Fig. 7h).

Altogether, our results indicate that EPR overexpression modulates cell proliferation and epithelial/mesenchymal markers levels in breast cancer cells and restrains cell proliferation in transplanted mice.

## Discussion

Here we report on the initial functional characterization of the long intergenic noncoding RNA EPR well conserved among mammalian species and expressed in select epithelial tissues including differentiated luminal cells of human breast. The levels of EPR are rapidly downregulated by TGF-β/SMAD signaling in immortalized mammary gland cells and its sustained expression largely reshapes the transcriptome by inducing epithelial traits while preventing the acquisition of mesenchymal markers upon TGF-β treatment. Remarkably, EPR overexpression enhances the levels of the cyclin-dependent kinase inhibitor CDKN1A and strongly reduces cell proliferation in both immortalized and transformed mammary gland cells as well as in transplanted mice.

EPR is almost equally distributed in chromatin, nucleoplasm and cytoplasm and the cytoplasmic component associates with polysomes where a small peptide (EPRp) is translated. EPRp interaction with a variety of cytoskeletal and junctional proteins accounts for its junctional localization. However, the analysis of the phenotype that we observed in cells overexpressing EPR mutants unable to originate the peptide clearly indicates that the vast majority of gene expression changes that we describe here are independent of the peptide biogenesis.

In this report, we investigated how the lncRNA molecule per se controls gene expression and we focused our studies on the EPR-dependent regulation of CDKN1A that functions as both a sensor and an effector of multiple antiproliferative signals and promotes cell cycle arrest in response to TGF-β[26]. In NMuMG cells, TGF-β induces an early wave of *Cdkn1a* expression due, in part, to an increased SMAD complex-dependent gene transcription while a prolonged treatment causes the return of *Cdkn1a* levels to the baseline. Our data suggest that *Cdkn1a* promoter-bound EPR recruits SMAD3 molecules—that accumulate into the nucleus upon TGF-β treatment for 1 h—to induce rapid gene transcription. In parallel, EPR interacts with KHSRP limiting its association with *Cdkn1a* mRNA and this results in the transcript stabilization. We propose that EPR downregulation upon 6 h of TGF-β treatment causes SMAD3 dismissal from *Cdkn1a* promoter that results in a return of *Cdkn1a* transcription to basal levels and, in parallel, enables KHSRP to destabilize the *Cdkn1a* transcript . Our data suggest that EPR-regulated molecular events shape the rapid wave of *Cdkn1a* expression in response to TGF-β. The evidence that CDKN1A is more abundant in cells overexpressing EPR than in mock cells (independently of any treatment with TGF-β) allows us to hypothesize that overexpressed EPR is able to recruit SMAD3 molecules already present in cell nuclei to the *Cdkn1a* promoter region and, possibly, to distal enhancers as well as to block KHSRP-induced *Cdkn1* mRNA degradation.

Our data indicate that EPR couples *Cdkn1a* transcriptional regulation with mRNA decay control. Indeed, the integration of transcription and mRNA decay provides a kinetic boost to a series of processes that would be otherwise slower and less efficient. This report strengthen the idea that coupling transcription to mRNA decay enables cells to rapidly modulate waves of gene

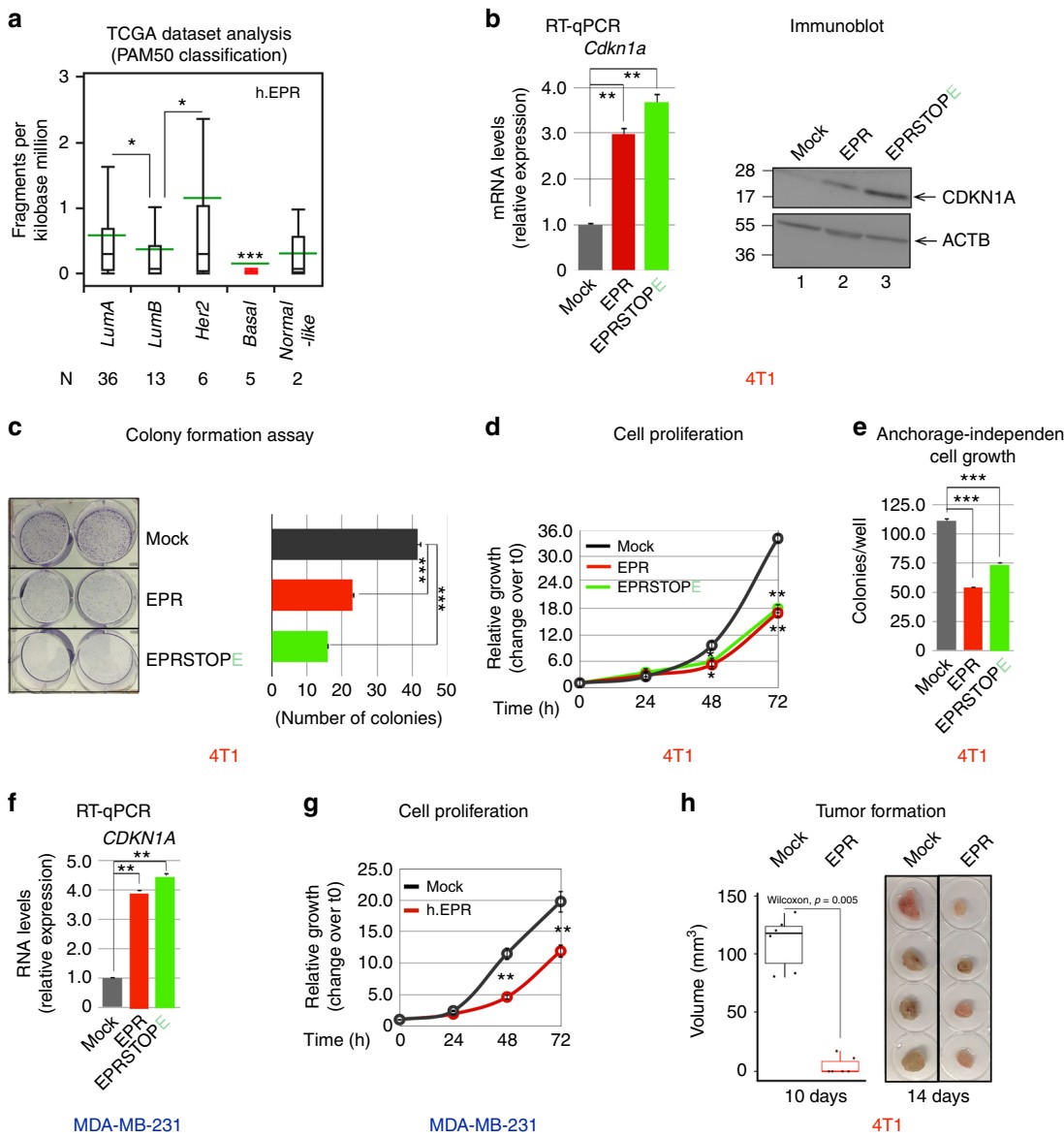

**Fig. 6** Antiproliferative effect of EPR when expressed in transformed mammary gland cells. **a** Box plot shows the expression of h.EPR in the TCGA Breast Cancer (BRCA) dataset annotated according to PAM50 molecular subtype classification. The number of samples in each subtype is presented. Asterisks mark significant values (Wilcoxon's test; *p < 0.05, ***p < 0.001). **b** qRT-PCR analysis (left panel) and immunoblot analysis (right panel) of Cdkn1a expression in either mock, EPR- or EPRSTOPE-overexpressing 4T1 cells. **c** Either mock, EPR- or EPRSTOPE-overexpressing 4T1 cells were seeded at low density and colony-formation assays were performed after 4 days. A representative plate is shown. The numbers are averages (±SEM) of four independent experiments performed in duplicate. Statistical significance: ***p < 0.00001 (Student's t test). **d** Cell proliferation analysis of either mock, EPR- or EPRSTOPE-overexpressing 4T1 cells. **e** Cells were cultured in soft agar for 21 days and phase contrast micrographs were taken at ×10 magnification. The values are averages (±SEM) of four independent experiments performed in triplicate. Statistical significance: ***p < 0.00001 (Student's t test). **f** qRT-PCR analysis of CDKN1A mRNA levels in transfected MDA-MB-231 cells. **g** Cell proliferation analysis of either mock or h.EPR-overexpressing MDA-MB-231 cells. **h** Tumor volume (n = 6 tumors/group) was measured by digital caliper assessment 10 days after injection of either mock or EPR-expressing 4T1 cells in BALB/c mice. Box plot analysis of tumor volume is shown (left). Box plots display summaries of the data distribution: the lower whisker is the minimum, the lower box edge is first quartile, the middle line is the median, the upper box edge is the third quartile, and the upper whisker the maximum value. Data were analyzed in R version 3.4.3, using the Wilcoxon Unpaired Test as implemented in "stat_compare_means" in "ggplot 2.2.1". Images of the tumors at the end of the experiment, 2 weeks after injection (right). The values of qRT-PCR experiments shown are averages (±SEM) of three independent experiments performed in triplicate. Statistical significance: **p < 0.001 (Student's t test). The values of cell proliferation experiments (panels **d** and **g**) are averages (±SEM) of three independent experiments performed in triplicate. Statistical significance: *p < 0.01, **p < 0.001 (Student's t test)

expression in response to a variety of stimuli thus achieving optimal mRNA homeostasis[30,31].

Since the decay of mature mRNAs is generally thought to occur in the cytoplasm, our finding that spliced and polyadenylated *Cdkn1a* mRNA is abundant in the nucleus of NMuMG cells, where it undergoes regulated decay, might be somehow surprising. However, the possibility that mature mRNAs accumulate in the nuclei of mammalian cells has been described[38,39] and a previous report has suggested that *Cdkn1a* mRNA undergoes degradation by the nuclear exosome in mammalian cells[40]. Further, a report about mature RNA degradation controlled by nuclear histone acetyltransferases and deacetylases provocatively

suggested the possibility that mRNA degradation pathways might operate also in the nucleus[41].

The TGF-β cytostatic program in epithelial cells involves, among other molecular events, the induction of *Cdkn1a* and *Cdkn2b* but cancer cells utilize any opportunity to circumvent TGF-β ability to inhibit cell proliferation. Inactivating mutations in the TGF-β Type-II receptors and SMAD4 have been described in tumors even though cancer cells can lose the cytostatic responsiveness due to defects downstream to SMAD factors[42,43]. We hypothesize that the absence of EPR/h.EPR, that occurs in breast cancer cell lines and in certain breast cancers, may contribute to the loss of TGF-β ability to restrain cell proliferation while may enable the cytokine to sustain their carcinogenic potential.

The notion that lncRNAs are devoid of coding potential has been recently challenged by reports demonstrating that translated short peptides are responsible for the biological functions of the respective lncRNA[22,23,44,45]. However, Yu et al.[46] reported that linc-RAM, an lncRNA that per se enhances myogenic differentiation by interacting with MYOD, is the transcript encoding myoregulin, a small peptide previously reported as a mediator of muscle performance through inhibition of the pump activity of SERCA[44]. In this case, RNA and peptide functions cooperate in modulating muscle physiology. Intriguingly, under our model the lncRNA per se is responsible for most of the gene expression changes while the peptide might be responsible for specific functions related to its cytoskeletal/junctional localization. We hypothesize that the peptide could participate in multiprotein complexes serving as permeability barriers and/or it could be implicated in the establishment of apico-basal polarity as well as in the transduction of signals to the cell interior. Further studies will be needed to clarify the specific functions of the peptide but we can predict that EPR and the peptide might synergize in executing epithelial cell-specific programs.

EPR was discovered as a highly regulated lncRNA in NMuMG mammary gland cells and, by exploiting the TGF-β-responsiveness of these cells, we were able to clarify some details of its molecular functions. How EPR influences normal physiology and disease in tissues undergoing repeated rounds of proliferation/differentiation, such as the gastrointestinal tract where it is highly expressed, will represent an important area of future research.

## Methods

**Cell lines**. Murine immortalized NMuMG cells (ATCC, no. CRL-1636) were cultured in Dulbecco's Modified Eagle Medium (DMEM) plus 10% Fetal Bovine Serum (FBS) and 10 µg ml$^{-1}$ bovine insulin (Sigma-Aldrich), 4T1 mouse mammary gland cancer cells (obtained from ATCC, no. CRL-2539) were cultured in DMEM/F12 plus 10% FBS, human mammary gland adenocarcinoma cells MDA-MB-231 (obtained from DSMZ, Germany, through Dr. G. Fronza, authenticated by STR DNA profiling) were cultured in DMEM plus 5% FBS, and human HEK-293 cells (ATCC, no. CRL-1573) were cultured in DMEM plus 10% FBS. HEK-293 cells were used to verify transient expression of FLAG-tagged EPRp based on their highly efficient transfectability. NMuMG cells were maintained in DMEM supplemented with 2% for 16 h prior to the addition of 1–10 ng ml$^{-1}$ human recombinant TGF-β1 purchased from R&D Systems. All cell lines were tested for mycoplasma contamination and resulted negative. SB431542 compound was purchased from Sigma-Aldrich, dissolved in Dimethyl sulfoxide (DMSO), and used at a 1 µM concentration. Cycloheximide, dissolved in DMSO, was purchased from Sigma-Aldrich and used at a 5 µM concentration.

**Antibodies and immunoblots**. Anti-EPRp polyclonal rabbit antibody was generated by injecting rabbits with recombinant purified EPR expressed in *E. Coli* using the pQE-EPR at Cambridge Research Biochemicals (Billingham, Cleveland, UK). Anti-CDH1 goat polyclonal antibody (sc-31020, used at 1:500 final dilution), anti-CDKN1A mouse monoclonal antibody (sc-6246, used at 1:200 final dilution), and anti-HDAC1 rabbit polyclonal antibody (sc-7872, used at 1:500 final dilution) were from Santa Cruz; anti-TJP1 rabbit polyclonal antibody (ab96587, used at 1:100 final dilution), anti-SMAD3 rabbit polyclonal antibody (ChIP grade ab28379), anti-GFP rabbit polyclonal antibody (ChIP grade ab90, used at 1:200 final dilution) were

from Abcam; mouse monoclonal anti-FLAG (F1804, used at 1:500 final dilution), mouse monoclonal anti-TUBA (DM1, used at 1:1000 final dilution) and mouse monoclonal anti-ACTB (AC-74, used at 1:30,000 final dilution) were from Sigma-Aldrich. Mouse monoclonal anti-RNA Polymerase II (clone CTD4H8) and rabbit polyclonal antibody to H3K27me3 (CS200603) were from Millipore. Rabbit polyclonal anti-CGN serum (C532, used at 1:5000 final dilution) was raised against a purified recombinant 50 kDa C-terminal fragment of chicken cingulin as well as anti-CGNL1 rabbit polyclonal antibody (20893, used at 1:100 final dilution) were raised at the University of Geneve. Images of the uncropped and unprocessed scans of the most important Immunoblots are presented in Supplementary Fig. 8.

**Plasmids**. Plasmid EPR was obtained by inserting the sequence from nucleotide 1 to 1487 of murine BC030870 into pBICEP-CMV-2 vector (Sigma-Aldrich); plasmid h.EPR was obtained by inserting the sequence from nucleotide 4 to 1126 of human LINC01207 into pBICEP-CMV-2 vector; plasmids EPRSTOPE and EPR-STOPM were obtained by Site-Directed Mutagenesis of plasmid EPR using the QuikChange II mutagenesis kit (Agilent Technologies) and the oligonucleotides 5′—CACCGTTAGTCTTCCATGTAGCTACCATTC—3′ and 5′—CACCGT-TAGTCTTCCTAGTAGCTACCATTC—3′, respectively. Plasmids EPR-FLAG and EPRSTOPE-FLAG were generated by inserting the sequence from nucleotide 1 to 560 of murine BC030870 obtained by PCR and Flagged at its 3′ (either wild-type or mutagenized as above) into pIRES1*hyg* vector. Plasmids GFP-mouse cortactin (#26722, CTTN-GFP) and CMV-GFP-human NMHC II-A (#11347, MYH9-GFP) were obtained from Addgene. Plasmid pQE-EPR was obtained by inserting the sequence from nucleotide 345 to 560 of murine BC030870 into pQE-30 vector (Qiagen).

For inserts obtained by RT-PCR, the Pfu DNA Polymerase (Promega) was used. The inserts cloned in all constructs were sequenced on both strands (BMR Genomics, Padova, Italy).

**Cell transfections**. NMuMG, 4T1, and MDA-MB-231 cells were transfected with Lipofectamine 2000 (ThermoFischer) while HEK-293 cells were transfected with Attractene transfection Reagent (Qiagen). NMuMG, 4T1, and MDA-MB-231 cells stably transfected with recombinant pBICEP-CMV-2-based vectors were maintained in selective medium containing 800, 350, and 750 µg ml$^{-1}$ G418 (Sigma-Aldrich), respectively. NMuMG cells stably transfected with recombinant pIRE-S1*hyg*-based vectors were maintained in selective medium containing 600 µg ml$^{-1}$ Hygromycin B (Sigma-Aldrich). Specific mock cells were generated (for every cell line and every plasmid backbone) by transfecting the corresponding empty vector in each cell type. Mock cells were subjected to a selection procedure identical to the other transfectants. siRNAs utilized to knockdown murine EPR (5′—GAG-CAAAAGAGAAUGCUUA—3′) were purchased from Thermo Fisher. Stable KHSRP knockdown in NMuMG cells was obtained using previously described silencing sequences and pSuper-Neo (Oligoengine) according to the manufacturer's instructions[15]. The adenoviral vectors pAdCMVnull (AdNull) and pAdKHSRP (full-length human KHSRP cDNA cloned into an Adenoviral-Type 5 backbone) were purchased from Vector Biolabs[15].

**Scratch wound closure assay**. Either mock or EPR-overexpressing NMuMG cells were cultured in six-wells plates up to confluence and pretreated for 2 h with 5 µg ml$^{-1}$ Mitomycin C (Sigma-Aldrich). A wound was scratched into monolayers and cells were cultured for up to 48 h in the presence of 5 µg ml$^{-1}$ Mitomycin C. Images were taken using an Olympus CKX41 microscope and analyzed using the ImageJ 1.49r package (http://imagej.nih.gov/ij/index.html). Average distance of wound obtained from six microscopic fields was used for the calculation of percent wound healed. Experiments were performed three times in triplicate.

**Immunofluorescence**. Either mock NMuMG cells or stable transfectants overexpressing either EPR-FLAG or EPRSTOPE-FLAG were plated on glass coverslips in 24-well plates (60,000 cells/well). Immunofluorescence was carried out 2 days after plating essentially as reported in ref. [47]. Rabbit polyclonal anti-cingulin antiserum (C532) was used at a 1:5000 dilution while anti-FLAG antibody (F1804, Sigma) was used at a 1:500 dilution. Secondary antibodies were diluted in IF buffer and incubated for 30 min at 37 °C, Alexa488 anti-rabbit (711-545-152, Jackson Laboratory) dilution 1:400, Cy3 anti-mouse (715-165-151, Jackson Laboratory) dilution 1/400. Pictures were taken using a Zeiss Axiophot widefield fluorescent microscope (X-Cite 120Q mercury lamp light source, Excelitas Technologies; retiga EXi, cooled mono 12-bit, Qimaging camera; ×63 oil objective; Openlab software). Images were imported into ImageJ to split and merge channels, cropped and adjusted for resolution and for intensity level range using Photoshop (scale bar = 10 µm).

**Orthotopic 4T1 injection in BALB/c mice**. BALB/c 8–10-week-old female mice (Envigo) were anesthetized using 100 mg kg$^{-1}$ ketamine and 10 mg kg$^{-1}$ xylazine intra peritoneal. Eye lubricant was applied, hair around the abdominal and inguinal fat pads were trimmed and the skin was sterilized. With the aid of magnifying surgical loupes, a small incision of less than 3 mm was made externally and caudally to the fourth nipple with the tip of micro-dissecting scissors. The fourth mammary gland fat pad below was located and 100 µl of a suspension of either

mock or EPR-expressing 4T1 cell were injected. Successful injection is confirmed by the swelling of the tissue. The incision was then sutured. All procedures involving animals have been approved by the Institutional Animal Welfare Body (O.P.B.A.) and complied with the national current ethical regulations regarding the protection of animals used for scientific purpose (D. Lvo, March 4, 2014, n. 26, legislative transposition of Directive 2010/63/EU of the European Parliament and of the Council of September 22, 2010 on the protection of animals used for scientific purposes). Tumor length and width were measured using a digital caliper at day 10 post injection and tumor volume was calculated using the formula: volume = (length × (width)$^2$/2). Mice were euthanized after 2 weeks and tumor masses were removed, weighted and photographed.

**Electrophoretic mobility shift assay.** Electrophoretic mobility shift assays (EMSA) were performed utilizing purified recombinant proteins that were incubated at room temperature for 20 min in a RNA-binding buffer (20 µl) containing 10 mM 4-(2-Hydroxyethyl)-1-piperazineethanesulfonic acid (HEPES) (pH 7.6), 3 mM MgCl$_2$, 100 mM KCl, 2 mM Dithiothreitol (DTT), 5% glycerol, 0.5% NP-40, yeast RNA (1 µg), and heparin (1 µg). The labeled RNA was transcribed using Sp6 polymerase from a template generated by inserting into pCY vector[48–50], a PCR product corresponding to nucleotides from 276 to 407 of murine BC030870.

**RNA isolation from cytoplasm, nucleoplasm, and chromatin.** We followed the protocol recently published by Corey and coworkers[51] starting from 10 × 10$^7$ cells. Both cytoplasmic and nucleoplasmic RNAs were precipitated and washed with ice-cold 70% (vol/vol) ethanol prior to be dissolved in QIAzol Lysis Reagent (Qiagen) while the chromatin pellets were immediately dissolved in QIAzol. Ten microliters of 0.5 M EDTA was added to all the samples in QIAzol that were heated to 65 °C with vortexing until dissolved (~10 min). The preparation of RNA was continued as described below. In parallel to RNA, protein extracts were prepared as described by Corey and coworkers[51].

**qRT-PCR, analysis of nascent transcripts, and mRNA decay.** Total RNA was isolated using either the miRNeasy mini kit or QIAzol (Qiagen) and retro-transcribed (50–100 ng) using Transcriptor Reverse Transcriptase (Roche) and random hexamers in most experiments according to the manufacturers' instructions. In order to verify if EPR is polyadenylated, qPCR amplification was performed using as template the product of reverse transcription reactions performed with oligo-dT (that pairs with the poly-A tail). Quantitative PCR was performed using the Precision 2× QPCR master mix (Primer Design), and the Realplex II Mastercycler (Eppendorf) according to the manufacturers' instructions. The sequence-specific primers utilized for PCR reactions are listed in Supplementary Data 3. In order to analyze gene expression changes among the pool of nascent mRNAs, we adopted the Click-iT Nascent RNA Capture kit (ThermoFischer) and performed experiments according to the manufacturer's instructions. NMuMg cells were pulsed with 0.5 mM 5-ethynyl Uridine (EU) for 1 h. In order to analyze mRNA decay we either blocked transcription by treating cells with 100 µM 5,6-Dichlorobenzimidazole 1-β-D-ribofuranoside (DRB, Sigma-Aldrich) and isolating total RNA at different intervals of times or performing EU labeling-based pulse chase experiments labeling cells with 0.2 mM EU for 16 h, removing the culture medium, and chasing cells for 1 h. RNA was prepared, clicked, retrotranscribed, and analyzed by qRT-PCR according to Click-iT Nascent RNA Capture kit instructions.

**Ribonucleoprotein complexes immunoprecipitation (RIP) assays.** Briefly, total cell lysates were immunoprecipitated with Dynabeads (Thermo Fisher) coated with protein A/protein G and precoupled to specific antibodies at 4 °C overnight. Pellets were washed three times with a buffer containing 50 mM Tris-HCl (pH 8.0), 150 mM NaCl, 0.5% Triton X-100, 1× Complete (Roche)[17]. Total RNA was prepared from immunocomplexes using the QIAzol Lysis Reagent, retrotranscribed, and amplified by qPCR as described above. The primer sequences are detailed in Supplementary Data 3.

**Protein identification by mass spectrometry (MS) analysis.** Total extract from either mock or EPR-FLAG (10 × 10$^7$ cells) were immunoprecipitated using anti-FLAG antibody-coupled Dynabeads. Immunoprecipitated material was analyzed by SDS-PAGE followed by silver staining. Protein identification was performed as a service at the Functional Proteomic Unit of IFOM (Milano, Italy; Drs. Angela Cattaneo and Angela Bachi). Bands of interest from SDS-PAGE were excised from gels, reduced, alkylated and digested overnight with bovine trypsin (Roche, Milan, Italy), as described[52]. One microliter aliquots of the supernatant were used for mass analysis using the dried droplet technique and α-cyano-4-hydroxycinnamic acid as matrix. Mass spectra were obtained on a MALDI–TOF Voyager-DE STR mass spectrometer (Applied Biosystem). Alternatively, acidic and basic peptide extraction from gel pieces after tryptic digestion was performed and the resulting peptide mixtures subjected to a single desalting/concentration step before MS analysis over Zip-TipC18 (Millipore Corporation). Spectra were internally calibrated using trypsin autolysis products and processed via Data Explorer software. Proteins were unambiguously identified by searching a comprehensive nonredundant protein database of the National Center for Biotechnology Information (NCBI, https://

www.ncbi.nlm.nih.gov/) and the Mass Spectrometry protein sequence DataBase (MSDB, http://msdn.microsoft.com/en-us/library/ms187112.aspx), selected by default using in-house software programs ProFound v4.10.5 and Mascot v1.9.00, respectively. Protein identifications were accepted if they could be established at greater than 99.0% probability and contained at least three identified peptides. Protein probabilities were assigned by the Protein Prophet algorithm[53]. Proteins that contained similar peptides and could not be differentiated based on MS/MS analysis alone were grouped to satisfy the principles of parsimony.

**RNA deep-sequencing (RNA-Seq).** High-quality RNA was extracted from either mock, EPR-, or EPRSTOPE- overexpressing NMuMG cells (biological triplicates for each experimental condition), and a total of nine libraries were prepared using standard Illumina TrueSeq SBS PE 200 cycles protocol and sequenced on HiSeq2500. Image analysis and base calling were performed using the HiSeq Control Software and RTA component from Illumina. This approach yielded between 68 and 77 millions of reads that were further processed.

**Analysis of h.EPR (LINC01207) expression in human samples.** Meta-analysis of RNA-Seq data of h.EPR in normal samples was performed by searching for h.EPR expression in different subpopulations of FACS-sorted normal breast cells[21] and in different human organs through either the Expression Atlas (https://www.ebi.ac.uk/gxa/home) or the GEPIA web server (http://gepia.cancer-pku.cn). h.EPR expression in breast cancer samples was analyzed using TCGA data, deriving PAM50 [37].

**Chromatin isolation by RNA purification (ChIRP).** Chromatin isolation by RNA purification (ChIRP) was performed according to the protocol published by Chu et al.[54] with minor modifications. Briefly, 2.5 × 10$^7$ NMuMG cells were crosslinked in 20 ml of 1% glutaraldehyde in PBS at room temperature for 10 min on an end-to-end rotator. After glutaraldehyde quenching and repeated washes, cell pellets were weighted and resuspended in 1.0 ml of complete Lysis Buffer (50 mM Tris-Cl pH 7.0, 10 mM EDTA, 1% SDS, 1× Complete, 500U RNAse inhibitor) per each 100 mg of cell pellet. Cell suspensions were sonicated for 90 min (power set to 70%) and the sonicated cell lysate was centrifuged at 16,100 × g at 4 °C for 10 min. Lysates were divided into two 1 ml aliquots, transferred into polypropylene tubes, mixed with 2 ml Complete Hybridization Buffer (750 mM NaCl, 1% SDS, 50 mM Tris-Cl, pH 7.0, 1 mM EDTA, 15% formamide, 1× Complete, 1000 U RNAse Inhibitor) and hybridized with 1 µl (100 pmol) of either EVEN or ODD pools of 20-mer 3′ Bio-TEG DNA oligonucleotides designed with single-molecule FISH online designer (Stellaris) (see Supplementary Data 3), respectively. Hybridization was carried out at 37 °C for 4 h under continuous shaking. Seventy microliters of prewashed C-1 magnetic beads (ThermoFisher) were added to each hybridization mixture for 30 min at 37 °C under continuous shaking. Beads were immobilized and washed four times for 5 min at 37 °C with shaking (wash buffer: 2× NaCl and Sodium citrate (SSC), 0.5% SDS, 1× Complete). While one aliquot (10% of the material) was utilized for RNA extraction, the remaining 90% was subject to DNA purification by incubating two times each bead pellet with 150 µl Complete DNA Elution Buffer (50 mM NaHCO$_3$, 1% SDS, 25 µg ml$^{-1}$ RNAseA, 100 U ml$^{-1}$ RNAseH) for 30 min at 37 °C with shaking. Eluted DNA was incubated with Proteinase K (1 mg ml$^{-1}$ final dilution) for 45 min at 50 °C with shaking, extracted with Phenol/Chloroform/Isoamylalchool, ethanol-precipitated, and aliquots were analyzed by qPCR.

**ChIP-qPCR.** ChIP experiments were performed according to the protocol published by Ghisletti et al.[55]. Briefly, ChIP lysates were generated from 40 × 10$^6$ cells. Each lysate was immunoprecipitated with 10 µg of anti-Pol II, anti-H3K27me3, and anti-SMAD3 antibodies (and the corresponding control IgG).

Antibodies were prebound overnight to 100 µl of A/G protein-coupled paramagnetic beads (ThermoFisher) in PBS/BSA 0.5%. Beads were then added to lysates (the preclearing step was omitted), and incubation was allowed to proceed overnight. Beads were washed six times in a modified RIPA buffer (50 mM HEPES (pH 7.6), 500 mM LiCl, 1 mM EDTA, 1% NP-40, and 0.7% Na-deoxycholate) and once in TE containing 50 mM NaCl. DNA was eluted in TE containing 2% SDS and crosslinks reversed by incubation overnight at 65 °C. DNA was then purified by Qiaquick columns (Qiagen) and quantified with PicoGreen (ThermoFisher).

**Sucrose-gradient fractionation and polysome profiling.** Experiments were performed as described[56]. NMuMG cells (~70% confluence) were treated with cycloheximide (0.1 mg ml$^{-1}$) for 5 min at 37 °C, washed twice with PBS supplemented by 0.01 mg ml$^{-1}$ cycloheximide, scraped in PBS 1× with 0.01 mg ml$^{-1}$ cycloheximide, pelleted by centrifugation, lysed in 500 µl of ice-cold Lysis Buffer (Salt Solution 1×, 1% Triton-X100, 1% NaDeoxycholate, 0.2 U µl$^{-1}$ RNase Inhibitor, 1 mM DTT, 0.01 mg ml$^{-1}$ cycloheximide), centrifuged for 5 min at 16,000 × g at 4 °C, and supernatants were loaded onto sucrose gradients. One milliliter fractions were collected monitoring the absorbance at 260 nm using a Density Gradient Fractionation System by Teledyne ISCO with sensitivity set to 0.2. Using the profile of the 260 nm absorbance, fractions corresponding to free ribosomal subunits (40S and 60S) and monosomes (80S, considered as not translating), separately from fractions corresponding to light polysomes (2–5 ribosomes) and

heavy polysomes (>6 ribosomes) were pooled together and processed for RNA extraction and RNA was quantified by Nanodrop (ThermoFisher).

**Cell cycle analyses by flow cytometer**. NMuMg cells (either mock or EPR- or EPRSTOPE-transfected) were seeded in six-well plates. For the analysis by the Cycletest™ Plus DNA Kit (BD Medical Technology), cells are detached by trypsinization and centrifuged in Eppendorf tubes at $300 \times g$ for 5 min at room temperature. Supernatant is removed and the pellet is resuspended in 1 ml 1× PBS followed by centrifugation. Cells are then resuspended in PBS and counted using Countess® Automated Cell Counter and the cell concentration is adjusted to $7 \times 10^5$ cells ml$^{-1}$ using the same buffer. The DNA staining procedure is performed using 0.5 ml of cell suspension ($7 \times 10^5$ cells). Cells are pelleted by centrifugation ($400 \times g$ for 5 min at RT). After carefully removing the supernatant, cells are mixed in Solution A (provided by the kit, containing trypsin in a spermine tetra-hydrochloride buffer for digestion of cell membranes and cytoskeleton), without using a vortex. Two hundred microliters of solution B (provided by the kit, containing trypsin inhibitor and ribonuclease A in citrate-stabilizing buffer, to inhibit the trypsin activity and to digest the RNA) is gently added and the sample is incubated for 10 min at RT, followed by the addition of 200 μl of cold solution C (provided by the kit, containing Propidium Iodide and spermine tetra-hydrochloride in citrate-stabilizing buffer). The sample is incubated in the dark and on ice for 10 min and then filtered by cell strainer caps and analyzed by flow cytometer (BD FACS Canto™). Data on at least 10,000 events for sample were processed using ModFit LT 4 software. The experiment was repeated two times.

To estimate more precisely the fraction of cells in S phase, the Click-iT™ Plus EdU Flow Cytometry Assay (Invitrogen) was used. EdU (10 mM stock in DMSO) was added directly to the culture medium at the 20 μM final concentration and incubated for 40 min. Cells were then harvested by trypsinization and washed using 3 ml of PBS containing 1% BSA. Pellets are resuspended in PBS+1% BSA, counted using Countess® Automated Cell Counter and $1.5 \times 10^6$ cells are transferred to flow tubes, washed again with 3 ml of PBS containing 1% BSA, pelleted by centrifugation followed by removal of the supernatant. Cells are resuspended in 100 μl of Click-iT™ fixative mixing well with a pipette and incubated for 15 min at RT in the dark. Cells are then washed as performed in the previous step, resuspended and incubated for 10 min in 100 μl of 1× Click-iT™ saponin-based reagent. Samples are then processed for the Click-iT™ reaction, preparing the Click-iT™ Plus reaction cocktail according to the manufacturer's guidelines, adding 0.5 ml of it to each sample, to reach a final volume of 600 μl containing $1.5 \times 10^6$ cells and incubating for 30 min at room temperature, in the dark. Cells are then washed once using 3 ml of 1× Click-iT™ saponin-based reagents, pelleted and resuspended in 600 μl of the same solution to which the propidium iodide staining solution is added to stain DNA. Propidium iodide solution contains 50 μg ml$^{-1}$ PI and 100 μg ml$^{-1}$ RNAse. Samples are then analyzed by flow cytometer (counting 20,000 events, BD FACS Canto™). As controls, cell aliquots incubated with EdU and processed by the same protocol, but skipping the Click-iT™ reaction or the PI staining, or both.

**G1 phase cell sorting**. Cells (mock, EPR, EPRSTOPE) were harvested by trypsinization when they reached ~90% confluence, washed once in 1× PBS and resuspended in DMEM without serum at a concentration of $10^6$ cells ml$^{-1}$. Hoechst 33342 (ThermoFisher Scientific) was added to the media at the concentration of 10 μg ml$^{-1}$ and cells were incubated for 1 h at 37 °C. Cells were then centrifuged to remove Hoechst-containing media and resuspended in 1× PBS.

Sorting was performed by BD Aria II™ cytometer (BD Bioscience) using a 100 μm nozzle and setting a gate on the population of cells in G1. At least 90 K events for every sample were sorted in 1× PBS at room temperature. After sorting, purity was assessed by re-running the samples. Sorted cells were pelleted and immediately stored at −80 °C. RNA was extracted and analyzed by qRT-PCR as described above.

**Cell proliferation analysis by high-content image analysis**. The proliferation of NMuMG cells (either mock or EPR- or EPRSTOPE-transfected) was quantified using Operetta High-Content Imaging System, acquiring images at different time points by digital phase contrast with a ×40 objective. Images were analyzed using Harmony® High Content Imaging and Analysis Software. Five hundred cells were seeded in 96-well plates in triplicates. Pictures were taken at different time points, by automatically acquiring eight fields for each well. Data were analyzed in Excel and plotted as average and standard deviations of replicates.

**Quantification of cell proliferation by crystal violet**. For some experiments cell proliferation was assessed by crystal violet staining. At the indicated time after plating, cells were fixed (10% formalin) and stained (0.1% crystal violet) with crystal violet solution. After two washes with water, crystal violet staining was measured by spectrophotometer at a wavelength of 590 nm.

**Clonogenic and anchorage-independent cell growth assays**. For the clonogenic assays, cells were plated in triplicate on six-well plates at 500 cells per well and left to grow for 4–6 days. Cells were fixed and stained with crystal violet solution. Anchorage-independent cell growth assays were assessed according to the protocol

published by Borowicz et al.[57] with minor modifications. Briefly, 2500 cells were seeded in 0.3% top agar in complete medium and placed on a layer of 0.5% of bottom agar in 12-well plates. Each cell line was seeded in sextuplicates and fed every 3 days. After 21 days cells were colored with crystal violet and photographs were taken.

**Quantification and statistical analysis of RNA-Seq**. Raw FASTQ reads were trimmed at the ends to remove low-quality calls with FASTX (http://hannonlab. cshl.edu/fastx_toolkit). Paired-end reads were aligned to indexed mm10 genome with STAR (v 2.3.0e_r291).

To quantify expression levels mapped reads were counted from BAM files with HTSeq counts version 1.2.1, in intersection-strict mode, feature type exon and id attribute gene_name against reference annotation Ensembl GRCm38.74.

**Quantitation of transcript differential expression analysis**. In addition to gene-level analysis with STAR-HTSeq, the transcript abundance was further re-estimated using an alignment-free approach based on Kallisto 0.43.1 software, using Gencode Mouse vM15 transcripts as reference.

Abundance files were imported in R.3.1.1 with TxImport.1.2.0 with option txOut =TRUE to quantify alternatively spliced transcripts. edgeR_3.16.5 and limma_3.30.13 were used to log2 transform transcripts count in Count Per Million (cpm). Only transcript with ≥1 cpm in at least three samples were retained. Cpm were transformed by library size and normalized by mean variance with limma-voom. Statistics and log-ratio were calculated with limma-eBayes, by fitting data to a single-factor linear-model with three different levels (mock, EPR, EPRSTOPE).

**Venn diagram and box plots**. We kept differentially expressed transcripts when the observed Bayesian statistic was significant (Benjamini and Hochberg corrected $p$ value <0.01; logFC >| 0.5 |). The functions limma-vennDiagram and pheatmap were used to cluster and visualize the significant genes. For box plots, summary statistics and plots were calculated and rendered with R software (version 3.5.0, https://www.R-project.org) through package ggpubr (https://CRAN.R-project.org/package=ggpubr) and ggplot2 (https://ggplot2.tidyverse.org). Data distributions and normality have been evaluated using the Shapiro−Wilk and Mann−Whitney tests for unpaired nonparametric data.

**Gene ontology and pathway enrichment**. Significant transcripts were summarized at gene level (tximport-summarizeToGene), annotated by Gene Ontology and enriched by statistically over-represented term with the EnrichR web-application using a nominal $p$ value (Student's $t$ test) threshold of $p < 0.01$. The EnrichR $p$ values refer to the Fisher Exact Test statistics, which is a proportion test that assumes a binomial distribution and independence for probability of any gene belonging to any set.

**Protein alignment**. Multiple alignment of mammalian EPR sequences was conducted by using the ClustalW2 package (https://www.ebi.ac.uk).

**Reporting summary**. Further information on research design is available in the Nature Research Reporting Summary linked to this article.

## Data availability
Raw data from RNA deep-sequencing analyses have been published on the GEO archive under the accession GSE113178. Human EPR expression in different subpopulations of FACS-sorted normal breast cells[21] and in different human organs was inferred through either the Expression Atlas (https://www.ebi.ac.uk/gxa/home) or the GEPIA web server (http://gepia.cancer-pku.cn). Proteins interacting with EPRp were unambiguously identified by searching a comprehensive nonredundant protein database of the National Center for Biotechnology Information (NCBI, http://www.ncbi.nlm.nih.gov/) and the Mass Spectrometry protein sequence DataBase (MSDB, http://msdn.microsoft.com/en-us/library/ms187112.aspx).

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

## Acknowledgements

We thank Pier Lorenzo Puri (SBP Medical Discovery Institute, La Jolla, CA) for discussions, Angela Cattaneo and Angela Bachi for discussions related to the MS analysis of EPRp interactors performed at the Proteomic Unit of IFOM (Milano), Sandro Poggi, Francesco Alessandrini, Federico Villa and Aldo Pagano (IRCCS San Martino) for sharing reagents and expertise, Claudia Bagni (University of Leuven), Paola Menichini and Gilberto Fronza (IRCCS San Martino) for sharing cells and reagents. This project has been supported, in part, by grants from the Associazione Italiana per la Ricerca sul Cancro (AIRC I.G. grant 10090 and 21541) to R.G., the Italian Ministry of Health with 5 × 1000 funds 2013 to P.B., and the Italian Ministry of Health with 5 × 1000 funds 2014 and 2015 to R.G. The work of G.B. was supported by the Italian Ministry of Health with 5 × 1000 funds. We wish to thank Michael Pancher of the High Throughput Screening facility, CIBIO, for assistance with high-content proliferation assay and Isabella Pesce of the Cell Analysis & Separation facility for assistance with the cell cycle analyses. We are grateful to people at TIB Mol.Biol. (Genova branch) for their professional assistance.

## Author contributions

P.B. and R.G. conceived the study and performed some experiments. M.R. performed most experiments. G.B. performed the bioinformatics analysis of the RNA-Seq data. D.R. performed polysome profiling, analyses of cell cycle distribution, and high-content image analysis. M.J.M. analyzed h.EPR expression in human samples. D.B. performed

bioinformatics analyses related to EPR peptide. A.F. and D.S. performed immuno-fluorescence experiments. M.P. produced the recombinant peptide EPRp and some recombinant plasmids. L.E. and M.C. performed the orthotopic transplants in mice. F.N., S. C. and A.I. analyzed and discussed the data obtained in their laboratories with P.B. and R.G. P.B. and R.G. wrote the manuscript. A.I. discussed the revision strategy with P.B. and R.G.

## Additional information

**Competing interests:** The authors declare no competing interests.

