## [Peer Review File · Nature Communications]

Reviewers' comments:

Reviewer #1, Expertise: Linc, cancer (Remarks to the Author):

The authors describe a previously uncharacterized gene, which they call linc-EPR, which is selectively expressed in epithelial tissues and is rapidly downregulated by TGF- β treatment. Linc-EPR modulates the expression of epithelial and mesenchymal markers and controls the acquisition of epithelial traits and cell proliferation both in cultured mammalian cells and in a mouse model of orthotopic transplantation. This gene is shown to encode a short peptide. The authors further show that linc-EPR regulates Cdkn1a expression through both transcription and mRNA decay in association with SMAD3 and KHSRP, respectively, in the context of TGF- β treatment. It is suggested that these effects are independent of the EPR peptide.

The topic of modes of action of lincRNAs and small peptides is of high interest, which makes the paper relevant for a large readership, and overall suitable for Nature Communications. The topic of bifunctional RNAs, which encode proteins and have lincRNA-like function is also intriguing, if proven convincingly. However, the data presented in the manuscript does not always corroborate the authors' conclusions. There are several concerns that should be addressed and suggested below are a number of experiments that will make it clearer whether the RNA or the encoded peptide are responsible for the reported effects of the EPR gene.

Major comments

1. The paper can be roughly divided into three parts – the first in which linc-EPR is described and some phenotypes are assigned, a second in which the EPR peptide is described and convincingly demonstrated to be a bona fide conserved protein, and then the third in which the authors argue that some functions of the linc-EPR gene, evolving around p21, are due to the activity of the RNA itself and are peptide-independent. The first two parts are convincing and fully worked out (and by themselves can suffice for a Nature Communications story in my eyes), but the third is less clear and less conclusive. Since the authors are mostly working in an over-expression setting, and compare linc-EPR over-expression to a “mock”, which I assume does not involve any plasmid transfection/over-expression (it is not explained what “mock” means), there is a concern that the peptide-unrelated effects may stem from the use of plasmid with a strong CMV promoter, which may cause some stress, or activate the PKR pathway (see e.g., <https://www.ncbi.nlm.nih.gov/pubmed/24475301>). Such stress may explain the shared effects of over-expression of linc-EPR and its versions where the peptide is mutated. Further, such stress is expected to potentially induce p21 levels. In some of the experiments in the last part of the paper, it is not currently clear whether the effects stem from the peptide, the potentially overexpression-related artefact concerns or, as stated, from an RNA-dependent activity (these concerns are related mostly to results in Figure 6c-I, but also 5b and 5e). These experiments need to compare linc-EPR with a proper control over-expression, EPR peptide over-expression, and with linc-EPRstope. Otherwise these parts of the manuscript are not conclusive and convoluted.

2. The synergism between TGFbeta or linc-EPR is not very convincing. Linc-EPR over-expression has a strong effect even without TGFbeta signalling (Fig. 1e), and TGFbeta has a strong effect on p21 without affecting the linc-EPR (Figure 4a). How do these fit into the authors' model? Figure 4d is also puzzling – if EPR is not induced by TGFbeta, is it expected to be strictly required for induction of p21? It is important to support this by FISH analysis of linc-EPR in TGF treated and untreated cells (see below). In addition, the cell migration assay (Supplementary Fig. 1j) should also be performed with TGF- β treatment as the mock cells seem to have an extensive migratory potential even in the absence of TGF- β .

Minor points

1. “log2 fold changes >|2.0|” (page 5) should be “|log2 fold change| >2.0”
2. Since the authors are describing here a previously uncharacterized gene, and show that it

encodes a peptide/protein, there is no reason to call it "linc-EPR", since by definition it is **not** a long noncoding RNA gene, but rather a protein-coding gene, which may have a 'moonlighting' lincRNA-like function. "EPR" would be a much better and less confusing nomenclature than "linc-EPR".

3. The authors should provide evidence that the linc-EPRSTOPM does not lead to production of EPR peptide.
4. There are no data or references supporting that KHSRP is predominantly nuclear in NMuMG cells (page 11).
5. Quality of the supplementary figures is very low, making it difficult to analyze some of the data provided within.
6. In Fig. 1e are differences following just TGF treatment in mock cells not significant? All differences should be evaluated and it should be clearly shown to what treatments the comparisons are being made.
7. An explanation as to what the asterisks in Fig. 2e represent should be provided in the figure legend.
8. Protein evidence (Western blot) should be provided to support the data in Fig. 3g and Supplementary Fig. 3d.
9. The authors refer to a ChIRP-Seq analysis performed in the laboratory that indicated the interaction of linc-EPR with the Cdkn1a promoter. These data should be provided. A FISH showing co-localization of linc-EPR with the Cdkn1a site of transcription is needed to support the idea that linc-EPR indeed interacts in trans with this promoter, as such interactions are still very rare and controversial.
10. The authors show in the supplementary Figs. 4f and 4g by qRT-PCR that the nuclear Cdkn1a is stabilized by linc-EPR overexpression. These data are used to substantiate the idea that it is the nuclear Cdkn1a mRNA that is destabilized upon KHSRP knockdown. However, this hypothesis should be validated by showing that the cytoplasmic Cdkn1a mRNA is less or not destabilized in the same experimental settings. These experiments should be provided since the levels of Cdkn1a mRNA in the cytoplasm are very similar to those found in the nucleus.
11. The authors state that the changes in Cdkn1a expression are independent of p53. Data supporting this claim should be provided, or it can be removed.

Reviewer #2, Expertise: epithelial, TGFbeta EMT (Remarks to the Author):

Rossi et al. (Gherzi) Nat Commun.

The authors identify a linc-RNA, named Linc-EPR, that is downregulated during TGF- β -induced EMT, is responsible for substantial changes in gene expression and promotes the epithelial phenotype. Focusing on one target gene, they show that Linc-EPR controls the transcription activation of the Cdkn1a gene (encoding p21Cip1/Waf1) and Cdkn1a mRNA degradation, which correlates with changes in proliferation. They propose that Linc-EPR antagonizes the TGF- β response, is under the control of TGF- β , and controls epithelial-mesenchymal transition and cell proliferation.

This is overall a good manuscript with some attractive observations. I may not agree with some conclusions, based on the data presented, and these conclusions need additional data as basis. I present these issues largely in order of appearance.

- Linc-EPR expression is seen as under the control of TGF- β signaling. I believe that this conclusion cannot be made since its decreased expression accompanies the initiation of EMT. Hence downregulation of Linc-EPR expression may result from the EMT process. Several experiments can resolve this issue. (1) Does TGF- β induce downregulation of Linc-EPR expression in cells that do not undergo EMT? (2) Does TGF- β induce downregulation of Linc-EPR expression, dependent on a

direct effect of Smads? This can be addressed in part by asking if the downregulation occurs in the absence of new protein synthesis (cycloheximide treatment), or by demonstrating Smad3 binding to the Linc-EPR promoter.

- As apparent from the title, the authors propose that Linc-EPR antagonizes TGF- β effects. This needs to be evaluated with some caution. Linc-EPR may antagonize the EMT program initiated by TGF- β , but not directly the response to TGF- β , and this may occur through its effect on cell proliferation (in part through the control of Cdkn1a expression). Are there other direct TGF- β /Smad target genes that are controlled directly by Linc-EPR? Does Linc-EPR control induction of Cdkn1a in response to other stimuli? And in cells that do not undergo EMT?

- Fig. 1b, S1g and corresponding text: The conclusion that Linc-EPR expression is restricted to epithelial cells is premature, based on the whole organ/tissue PCR data presented. All the organs/tissues analyzed by RT-PCR contain a mixture of cells. In situ hybridization data of tissue slices showing epithelial and other cells is needed to evaluate whether Linc-EPR expression is restricted to epithelial cells.

- Fig. 1f, S1h and other figure panels: The NMuMG cells that are not transfected with Linc-EPR do not look epithelial, yet should look epithelial. Additionally, Fig. 3c does not show detectable E-cadherin expression, whereas they should be epithelial and express E-cadherin. As the cells were apparently not treated with TGF- β , I am concerned that they are not behaving the way they should and have drifted toward a mesenchymal phenotype.

- The transcriptome analyses led the authors to conclude that a large number of genes are controlled by Linc-EPR. However, the cell cycle analyses in Fig. 3i show major changes in the fractions of cells in different stages of the cell cycle, which may be due in part to changes in Cdkn1a expression. Therefore the major changes in the transcriptome might result from the cell cycle changes and the relative changes in fractions of cells in different cell cycle stages, and not from Linc-EPR expression per se. This is an important issue, and might be addressed by evaluating cells that are in the same phase of the cell cycle.

- lines 249-252: The statement about p53 should be backed up by data. No data are shown.

- Fig. 5a: One should assume that most readers are not familiar with ChIRP-qPCR, and what even and odd primers are doing in this context is unclear. Please explain.

Point-by-point list of the revisions.

REVIEWER 1

The topic of modes of action of lincRNAs and small peptides is of high interest, which makes the paper relevant for a large readership, and overall suitable for Nature Communications. The topic of bifunctional RNAs, which encode proteins and have lincRNA-like function is also intriguing, if proven convincingly. However, the data presented in the manuscript does not always corroborate the authors' conclusions. There are several concerns that should be addressed and suggested below are a number of experiments that will make it clearer whether the RNA or the encoded peptide are responsible for the reported effects of the EPR gene.

Major comments

1. The paper can be roughly divided into three parts – the first in which linc-EPR is described and some phenotypes are assigned, a second in which the EPR peptide is described and convincingly demonstrated to be a bona fide conserved protein, and then the third in which the authors argue that some functions of the linc-EPR gene, evolving around p21, are due to the activity of the RNA itself and are peptide-independent. The first two parts are convincing and fully worked out (and by themselves can suffice for a Nature Communications story in my eyes), but the third is less clear and less conclusive. Since the authors are mostly working in an over-expression setting, and compare linc-EPR over-expression to a “mock”, which I assume does not involve any plasmid transfection/over-expression (it is not explained what “mock” means), there is a concern that the peptide-unrelated effects may stem from the use of plasmid with a strong CMV promoter, which may cause some stress, or

activate the PKR pathway (see e.g., <https://www.ncbi.nlm.nih.gov/pubmed/24475301>). Such stress may explain the shared effects of over-expression of linc-EPR and its versions where the peptide is mutated. Further, such stress is expected to potentially induce p21 levels. In some of the experiments in the last part of the paper, it is not currently clear whether the effects stem from the peptide, the potentially overexpression-related artefact concerns or, as stated, from an RNA-dependent activity (these concerns are related mostly to results in Figure 6c-I, but also 5b and 5e). These experiments need to compare linc-EPR with a proper control over-expression, EPR peptide over-expression, and with linc-EPRstope. Otherwise these parts of the manuscript are not conclusive and convoluted.

We are pleased to read that Reviewer 1 appreciated our effort to provide a comprehensive portrait of EPR biology and we have done our best to provide solid experimental evidence supporting the mechanistic functional model that we propose.

First of all we apologize for the confusion mainly created by the Methods section. We fully agree with the Reviewer that it is not correct to compare transfected cells with cells not subject to the transfection procedure. Indeed, we did not perform such comparison in our work but we recognize that we did not describe with sufficient clarity our experimental procedure. In this revised manuscript (Methods, Cell Transfections sub-section [page 31] and first sub-section of the Results [page 6]) we now clearly indicate that the transfectants that we generated in distinct mammary gland cell lines

(using plasmids with different backbones) were compared with “real” mock transfectants (i.e. all the EPR transfectants were always analyzed together with the respective mock cells transfected with the corresponding empty vector —thus including the same promoter— and subject to the same selection procedure).

Further, we have diligently worked to add experimental evidence to prove that the mutated EPR version unable to generate the peptide displays the majority of functions exerted by the non-mutated EPR. The new data are presented in Figure 4a, b; Figure 5b, e; Figure 6b-f; Supplementary Figure 5c; and in the new Supplementary Figure 3g. As for data presented in Panel h of Figure 6, the Italian regulatory organism controlling animal experimentation (conforming to the “Reduction” policy adopted in Europe, Directive 2010/63/EU of the European Parliament and of the Council of September 22, 2010 on the protection of animals used for scientific purposes), did not allow us to test the effect of EPRSTOPE-transfected 4T1 cells in BALB/c mice considering that all the results obtained in cell culture indicate an equal activity on cell proliferation of either EPR or EPRSTOPE when overexpressed in 4T1 cells.

Finally, in order to support the data presented in our manuscript, we would also like to share with the Reviewer an additional set of data obtained using NMuMG cells stably transfected with either a shorter versions of EPR (nt. 1-560 of the EPR sequence which include the ORF herein referred as to 5'+EPRp) or its derivative carrying a STOP mutation in the second codon (5'+EPRpSTOPE). Also in this case (please see the Figure below) both transfected plasmids yield superimposable results in terms of CDKN1A expression and cell proliferation. These experiments are part of an effort to define the EPR regions responsible for the effects described in this manuscript and will be part of a different study.

a. Immunoblot analysis of total cell extracts from either mock, 5'+EPRp-, and 5'+EPRpSTOPE- overexpressing NMuMG cells. (b) Proliferation analysis of either either mock, 5'+EPRp-, and 5'+EPRpSTOPE- overexpressing NMuMG cells.

2. The synergism between TGFbeta or linc-EPR is not very convincing. Linc-EPR over-expression has a strong effect even without TGFbeta signalling (Fig. 1e), and TGFbeta has a strong effect on p21 without affecting the linc-EPR (Figure 4a). How do these fit into the authors' model? Figure 4d is also puzzling – if EPR is not induced by TGFbeta, is it expected to be strictly required for induction of p21? It is important to support this by FISH analysis of linc-EPR in TGF treated and untreated cells (see below). In addition, the cell migration assay (Supplementary Fig. 1j) should also be performed with TGF-β treatment as the mock cells seem to have an extensive migratory potential even in the absence of TGF-β.

Also in this case, we apologize for the lack of clarity in our text that we have amended in this revised version in order to make clear the model that we propose (pages 14 and 15 of the Discussion section). Further, we added new ChIRP-qPCR experiments (see new Supplementary Figure 4b) showing that TGF-β treatment for 1 hour does not change the interaction of EPR with *Cdkn1a* promoter (as for the technique used to demonstrate direct interaction between EPR and the *Cdkn1a* promoter, please see our response to minor point # 9). The new results, together with the experimental data already included in the manuscript, allow us to propose the following model. TGF-β induces an early wave of *Cdkn1a* expression due, in part, to an increased SMAD complex-dependent gene transcription. A prolonged TGF-β treatment causes the return of *Cdkn1a* levels to the baseline. EPR, which is similarly bound to *Cdkn1a* promoter either in cells untreated or treated with TGF-β for 1 hour, recruits SMAD3 molecules when they accumulate into the nucleus upon treatment with the cytokine and this leads to rapid *Cdkn1a* gene transcription. Thus, we believe that the limiting step in the rapid transcriptional induction of *Cdkn1a* is represented by the enhanced availability of SMAD3 upon TGF-β treatment and its recruitment by EPR. In parallel, EPR interacts with KHSRP limiting its association with *Cdkn1a* mRNA and this results in the stabilization of the transcript. We propose that EPR down-regulation upon 6 hours of TGF-β treatment causes SMAD3 dismissal from *Cdkn1a* promoter that results in a return of *Cdkn1a* transcription to basal levels and, in parallel, enables KHSRP to interact with *Cdkn1a* mRNA and to destabilize it. We propose that EPR-regulated molecular events shape the rapid wave of *Cdkn1a* expression in response to TGF-β in NMuMG cells. The evidence that CDKN1A is abundant in cells overexpressing EPR even in the absence of TGF-β treatment allows us to hypothesize that overexpressed EPR is able to recruit SMAD3 molecules already present in cell nuclei to *Cdkn1a* promoter region and, possibly, to distal enhancers as well as to block KHSRP-induced *Cdkn1* mRNA degradation.

As for the migratory potential in the absence of any TGF-β treatment, NMuMG cells display the ability to migrate also in the absence of the cytokine (see Romagnoli et al., *Cancer Res.* 2012; Puppo et al., *Cell Rep.* 2016). However, TGF-β strongly enhances NMuMG migratory potential. We have now completed the data presented in the new Supplementary Figure 1k with the addition of assays performed in cells treated for 24 hours with TGF-β according to Reviewer's suggestion. For the

Reviewer's convenience, we also show in the Figure here below that a significant difference in the migratory potential of EPR-overexpressing NMuMG cells compared with mock cells is already evident after 8 hours of TGF- β treatment.

Scratch wound healing assays. Cultures were either treated with TGF- β (for 8 hours (+)) or left untreated (–). The percentage of the scratch gap area for each culture condition was plotted. Statistical significance: **p < 0.001 (Student's t test).

Minor points

1. “log2 fold changes >|2.0|” (page 5) should be “|log2 fold change| >2.0”

log2 fold changes >|2.0|” has been replaced with “|log2 fold change| >2.0.”

2. Since the authors are describing here a previously uncharacterized gene, and show that it encodes a peptide/protein, there is no reason to call it “linc-EPR”, since by definition it is *not* a long noncoding RNA gene, but rather a protein-coding gene, which may have a 'moonlighting' lincRNA-like function. "EPR" would be a much better and less confusing nomenclature than "linc-EPR".

We thank the Reviewer for her/his suggestion and we adopted the new nomenclature renaming the linc-EPR as EPR and the peptide as EPRp.

3. The authors should provide evidence that the linc-EPRSTOPM does not lead to production of EPR peptide.

We added a new Supplementary Figure 3c showing that the mutated EPRpSTOPM (STOP mutation in the first Methionine) is not expressed. Further, we show in this revised manuscript that the overexpression of EPRSTOPM affects gene expression similarly to EPRSTOPPE.

4. There are no data or references supporting that KHSRP is predominantly nuclear in NMuMG cells (page 11).

We are sorry for not being clear enough. We have previously reported that KHSRP is predominantly nuclear in NMuMG cells (Puppo, Bucci et al., *Cell Rep.*, 2016, cited as reference 15 in our

manuscript) and for Reviewer's convenience we also show here below Immunoblots demonstrating the nucleoplasmic and chromatin localization of KHSRP.

NMuMG cells were fractionated, protein extracts were prepared from cytoplasm, nucleoplasm, and chromatin and analyzed by Immunoblot. The indicated antibodies were used; the position of molecular mass markers is presented on the left.

5. Quality of the supplementary figures is very low, making it difficult to analyze some of the data provided within.

We are really sorry that the Reviewer had difficulties with some of our Figures. We are doing our best during the uploading procedure in order to make available Figures of the best possible quality to Reviewers.

6. In Fig. 1e are differences following just TGF treatment in mock cells not significant? All differences should be evaluated and it should be clearly shown to what treatments the comparisons are being made.

We apologize for the poor labeling of Figure 1e (and of other panels). We amended our mistake in the revised Figures 1e (and also 4a, 4f, 5f, 6b, 6e, and 6f).

7. An explanation as to what the asterisks in Fig. 2e represent should be provided in the figure legend.

In the legend to Figure 2e, we now indicate that Asterisks mark the position of immunoglobulin heavy and light chains.

8. Protein evidence (Western blot) should be provided to support the data in Fig. 3g and Supplementary Fig. 3d.

Immunoblots have been added to the revised Figure 3g and Supplementary Figure 3e (formerly 3d).

9. The authors refer to a ChIRP-Seq analysis performed in the laboratory that indicated the interaction of linc-EPR with the *Cdkn1a* promoter. These data should be provided. A FISH showing co-localization of linc-EPR with the *Cdkn1a* site of transcription is needed to support the idea that linc-EPR indeed interacts in trans with this promoter, as such interactions are still very rare and controversial.

We apologize for the lack of clarity of the original manuscript. The Chromatin Isolation by RNA Purification (ChIRP) experiments shown in Figure 5a and in the new Supplementary Figure 4b have been performed on the basis of an initial analysis of ChIRP-Seq datasets (currently under in-depth bioinformatics evaluation) that we undertook to precisely map the genomic binding sites of EPR. The preliminary analysis of the data allowed us to identify EPR binding sites in the promoter of *Cdkn1a* gene. This was validated by the ChIRP-qPCR experiments shown in Figure 5a. Although the complete ChIRP-Seq analysis will constitute the basis for additional studies, we provide for the Reviewer's convenience (in the Figure here below) a portrait of EPR interaction with *Cdkn1a* promoter as well as with the promoter of an additional EPR target gene that is currently under investigation in our laboratory (*Cdx2*, please see also our response to Reviewer 2).

As for the technique used in order to show the interaction of EPR with the *Cdkn1a* promoter, we would like to respectfully underline that we adopted ChIRP because, since its original discovery and application by Dr. Chang's laboratory, it has been recognized and validated as one of the methods of choice to map lincRNA interactions with DNA (as well as with protein factors). We hope that the Reviewer will appreciate our effort in setting up and optimizing this complex technique in NMuMG cells.

Screenshots of Integrative Genomics Viewer windows showing representative ChIRP-Seq data analysis.

10. The authors show in the supplementary Figs. 4f and 4g by qRT-PCR that the nuclear Cdkn1a is stabilized by linc-EPR overexpression. These data are used to substantiate the idea that it is the nuclear Cdkn1a mRNA that is destabilized upon KHSRP knockdown. However, this hypothesis should be validated by showing that the cytoplasmic Cdkn1a mRNA is less or not destabilized in the same experimental settings. These experiments should be provided since the levels of Cdkn1a mRNA in the cytoplasm are very similar to those found in the nucleus.

We thank the Reviewer for asking us to better clarify this point. The new Supplementary Figure 4g shows that cytoplasmic *Cdkn1a* mRNA is relatively stable in mock cells and its decay rate is not significantly affected by EPR overexpression.

11. The authors state that the changes in Cdkn1a expression are independent of p53. Data supporting this claim should be provided, or it can be removed.

Following Reviewer 1 and 2 suggestion, we removed the sentence regarding p53.

REVIEWER 2

This is overall a good manuscript with some attractive observations. I may not agree with some conclusions, based on the data presented, and these conclusions need additional data as basis. I present these issues largely in order of appearance.

- Linc-EPR expression is seen as under the control of TGF- β signaling. I believe that this conclusion cannot be made since its decreased expression accompanies the initiation of EMT. Hence downregulation of Linc-EPR expression may result from the EMT process. Several experiments can resolve this issue. (1) Does TGF- β induce downregulation of Linc-EPR expression in cells that do not undergo EMT? (2) Does TGF- β induce downregulation of Linc-EPR expression, dependent on a direct effect of Smads? This can be addressed in part by asking if the downregulation occurs in the absence of new protein synthesis (cycloheximide treatment), or by demonstrating Smad3 binding to the Linc-EPR promoter.

We greatly appreciate the overall positive comment of this Reviewer and her/his constructive criticisms that allowed us to improve our manuscript.

(1) We observed that down-regulation of EPR expression occurs in colon carcinoma cells LS180 upon TGF- β treatment also in the absence of any modulation of factors relevant during EMT. We show these data in the Figure below for the Reviewer's convenience.

qRT-PCR analysis of h.EPR and other transcripts in LS180 cells treated as indicated. Statistical significance: **p < 0.001 (Student's t test).

(2) Following Reviewer's suggestion, we include in this revised manuscript (Supplementary Figure 1e) new data showing that SMAD3 interacts with the EPR promoter region and that this interaction is modulated by TGF- β treatment. This observation suggests that EPR expression is under the direct control of the TGF- β /SMAD signaling pathway. Experiments aimed at investigating the detailed molecular mechanisms underlying EPR regulation by TGF- β are in progress in our laboratory and will be part of further studies.

- As apparent from the title, the authors propose that Linc-EPR antagonizes TGF-b effects. This needs to be evaluated with some caution. Linc-EPR may antagonize the EMT program initiated by TGF-b, but not directly the response to TGF-b, and this may occur through its effect on cell proliferation (in part through the control of Cdkn1a expression). Are there other direct TGF-b/Smad target genes that are controlled directly by Linc-EPR? Does Linc-EPR control induction of Cdkn1a in response to other stimuli? And in cells that do not undergo EMT?

We thank again the Reviewer for giving us the opportunity to better characterize some mechanistic aspects of EPR function.

— In the course of the analysis of ChIRP-Seq datasets (please see also our response to Reviewer 1) we have identified the gene encoding the transcription factor CDX2 as a direct target of TGF- β /SMAD signaling (panel a and b in the Figure below and Barros et al., *J. Pathol.*, 2008). Similarly to *Cdkn1a*, the expression of *Cdx2* is strongly enhanced by EPR overexpression (panel c) and ChIRP experiments revealed that EPR directly interacts with *Cdx2* promoter (panel d). Considering the important role of CDX2 in the intestine and the existence of some reports indicating that *Cdx2* and *Cdkn1a* belong to the same regulatory pathway (Bai et al., *Oncogene* 2003; Mari et al., *Cell Rep.*, 2014; Parveen et al., *Crit. Rev. Eukaryot. Gene Expr.*, 2016), we plan to investigate the relationships between EPR, *Cdkn1a*, and *Cdx2* in future studies using the gastrointestinal tract as a model.

(a) qRT-PCR analysis of *Cdx2* expression in NMuMG cells serum-starved (2% FBS, 16h) and either treated with TGF-β (10 ng/ml) for the indicated times or untreated (time 0). (b) Chromatin prepared from NMuMG cells serum-starved and either treated with TGF-β for the indicated times or untreated (control) was immunoprecipitated using either normal rabbit IgG (clgG) or affinity-purified anti-SMAD3 rabbit polyclonal antibody. The association of SMAD3 with *Cdx2* promoter (schematic on the top) was quantitated by qPCR using specific primers (indicated as arrowheads). (c) qRT-PCR analysis of *Cdx2* in either mock, EPR- or EPRSTOPE- expressing NMuMG cells. (d) ChIRP analyses performed using NMuMG cell lysates and either even or odd EPR probe sets. Both input and purified DNA were analyzed by qPCR using primers designed on *Rpl32* (negative control) or *Cdx2* promoters. Values are averages (±SEM) of three independent experiments performed in triplicate.

— We have found that the DNA damaging agent Doxorubicin downregulates EPR expression in a concentration-dependent manner and, in parallel, induces *Cdkn1a* expression in NMuMG cells (see figure below). Thus, in contrast to the TGF-β/SMAD signaling, this stimulus causes a reduction of EPR expression that correlates with an increase of *Cdkn1a* expression.

qRT-PCR analysis of EPR and *Cdkn1a* expression in NMuMG cells treated for 16 hours with the indicated concentrations of Doxorubicin or left untreated (0).

In conclusion, taking into account all the new experimental evidences included in the revised manuscript (with particular emphasis on data presented in Supplementary Figure 1e) as well data included in this letter, we believe that EPR expression/function is part of the TGF- β -SMAD signaling. However, we highly evaluate the Reviewer's concern and we have decided to change the title of the manuscript by removing the notion of "antagonism" between EPR and the TGF- β signaling that can lead to misunderstandings.

- Fig. 1b, S1g and corresponding text: The conclusion that Linc-EPR expression is restricted to epithelial cells is premature, based on the whole organ/tissue PCR data presented. All the organs/tissues analyzed by RT-PCR contain a mixture of cells. In situ hybridization data of tissue slices showing epithelial and other cells is needed to evaluate whether Linc-EPR expression is restricted to epithelial cells.

The Reviewer is perfectly right. We re-analyzed the RNA-Seq data derived from different subpopulations of normal breast cells isolated by FACS analysis from reduction mammoplasty specimens (presented in Figure 1d; Pellacani et al., *Cell Rep.*, 2016, reference 18) and we found that stromal cells display EPR expression below the detection levels. This data is now shown in the revised Figure 1d. Further, data derived from single cell analysis in mouse confirmed the exclusive expression of EPR in the luminal cells of mammary gland (<http://bis.zju.edu.cn/MCA/search.html>). Of course we cannot exclude that EPR is expressed also in non-epithelial cells of other tissues and, as a consequence, we have moderated our statements throughout the entire manuscript by eliminating the notion of "epithelial-restricted" EPR expression.

Fig. 1f, S1h and other figure panels: The NMuMG cells that are not transfected with Linc-EPR do not look epithelial, yet should look epithelial. Additionally, Fig. 3c does not show detectable E-cadherin expression, whereas they should be epithelial and express E-cadherin. As the cells were apparently not treated with TGF- β , I am concerned that they are not behaving the way they should and have drifted toward a mesenchymal phenotype.

In this revised version we provide new phase contrast microscopy images (new Figure 1g) showing more confluent cell cultures since we realized that the epithelial morphology of NMuMG cells can be less evident at lower confluence. Also, we substituted the anti-CDH1 immunoblot shown in Figure 3c with a new one (longer exposure) in which the CDH1 expression is evident also in mock cells.

We would also like to share with the Reviewer additional evidence that NMuMG cells used for this study (purchased at ATCC, no. CRL-1636, maintained in culture no longer than two weeks after thawing, and repeatedly frozen) display the expected epithelial characteristics. In the Figure below is shown an example of the preliminary characterization that has been performed before using the cells for the experiments. Epithelial markers such as Occludin and Afadin are properly expressed and localized.

NMuMG mock

Immunofluorescence analysis (using the indicated antibodies) of mock-transfected NMuMG cells cultured to confluence.

- The transcriptome analyses led the authors to conclude that a large number of genes are controlled by Lnc-EPR. However, the cell cycle analyses in Fig. 3i show major changes in the fractions of cells in different stages of the cell cycle, which may be due in part to changes in Cdkn1a expression. Therefore the major changes in the transcriptome might result from the cell cycle changes and the relative changes in fractions of cells in different cell cycle stages, and not from Lnc-EPR expression per se. This is an important issue, and might be addressed by evaluating cells that are in the same phase of the cell cycle.

We thank the Reviewer for her/his criticism that prompted us to include new data into our revised manuscripts. Data presented in the new Figure 3g indicate that the gene expression changes induced by either EPR or EPRSTOPE in G1-enriched cells are superimposable to those observed in the total cell population (please compare Supplementary Fig. 3g with Fig. 3b).

- lines 249-252: The statement about p53 should be backed up by data. No data are shown.

As requested also by Reviewer 1 we have deleted the statement about p53.

- Fig. 5a: One should assume that most readers are not familiar with ChIRP-qPCR, and what even and odd primers are doing in this context is unclear. Please explain.

We are sorry for the lack of clarity. As described in the original protocol from Dr. Chang's laboratory, we denominated Even and Odd two distinct sets of tiling biotinylated oligonucleotides that are used in order to isolate complexes between the lncRNAs and target genomic regions. The use of two distinct sets of tiling oligonucleotides contributes to obtain specificity in the procedure. We have added a schematic on the top of Figure 5a in order to make more clear the ChIRP procedure and the position of the oligonucleotides.

Reviewers' comments:

Reviewer #1 (Remarks to the Author):

The authors have addressed the concerns from the previous round of review in a satisfactory way, and the newly added data provide substantial support to their model. The only remaining minor comment (that the authors can be trusted to address by themselves) is to add as a last panel in the paper an illustration of the proposed model.

Reviewer #2 (Remarks to the Author):

Rossi et al. (Gherzi) Nat Commun.

The authors identify a linc-RNA, named linc-EPR, that is downregulated during TGF- β -induced EMT, is responsible for substantial changes in gene expression and promotes the epithelial phenotype. Focusing on one target gene, they show that linc-EPR controls the transcription activation of the *Cdkn1a* gene (encoding p21^{Cip1/Waf1}) and *Cdkn1a* mRNA degradation, which correlates with changes in proliferation. They propose that linc-EPR antagonizes the TGF- β response, is under the control of TGF- β , and controls epithelial-mesenchymal transition and cell proliferation.

As I mentioned in my review of the previous version, this is overall a good manuscript with attractive observations. I previously requested additional data to strengthen the conclusions, and feel that this was done only to some extent, and that, in some cases the authors preferred to weaken their conclusions rather than to strengthen their data. Furthermore, some figure panels are not sufficiently informative about controls, what was done and how the reader should evaluate the data. In providing my comments previously, I hoped that this manuscript would have been better revised, rather than just adding or changing the yellow highlighted sentences (and corresponding data panels). One question that still stands out, and that I steered towards in my previous comments, relates to the TGF- β -inducibility of this system. Based on what I see in the data and my experience, the EPR activity will be TGF- β dependent; however, the authors do not show convincing data in that regard. Indeed, the data without adding TGF- β seem to reflect a high level of autocrine TGF- β signaling. In other words, they often compare low with higher TGF- β signaling, rather than no TGF- β versus +TGF- β . Blocking (autocrine) TGF- β signaling is easily achieved using the T β RI kinase inhibitor SB431542, but this was not done. Comparing +SB431542 and +TGF- β would give much better results. Maybe I was not sufficiently clear on how to address the TGF- β dependence, but I feel reluctant to prescribe exactly what experiments need to be done and how to do things. Finally, in their rebuttal, the authors show me some data that would benefit the manuscript. Incorporating these into the manuscript would strengthen it. After all, I am reviewing it from the standpoint of a critical reader, and the reader does not consult the reviews.

I present the following issues largely in order of appearance.

- line 103: Fig. S1b and its legend are insufficiently informative. How can I see that TGF- β modulates the interaction of KHSRP with 67 lincRNAs?
- line 108: In Fig. S1c, what does P.I. stand for?
- line 108: In Fig. S1d, there is no control for GST-KHSRP. GST by itself (not shown) is normally not a good control for such experiments. GST fused to another protein should serve as a negative control.
- line 110: How do I know from looking at Fig. S1b that there is reduced interaction of EPR with KHSRP at 6 h?
- lines 111-113: The ChIP-qPCR in Fig. S1e is not convincing. Any controls? Also, the untreated sample shows very high Smad3 binding. Is this basal autocrine TGF- β signaling? If so, why does it decrease so much at 6 h? What is the level if you block TGF- β signaling using SB431542. These data do not look good, and do not allow for conclusions.

- As requested before, is the gene encoding EPR a direct target of TGF- β /Smad signaling? Is it induced (and then downregulated) in response to TGF- β + cycloheximide to block new protein synthesis? This should be a simple experiment.
- line 117: Why do the authors refer to Fig. S1g? What is shown? I cannot figure it out from the legend, nor do I understand it.
- line 117: Fig. 1c has no controls. What are the controls for the different compartments? If Neat1 and tRNA-Lys are seen as controls, then it looks to me that the fractionation was not good at all. The standard controls for subcellular fractionation are needed.
- lines 123-124: Any possibility to show that EPR is made in other epithelia? For example, colon, which is high in Fig. 1b. The luminal epithelial expression is interesting, but that result has only a very narrow scope. It would be nice to see more evidence that it is expressed in epithelial or differentiated epithelial cells.
- line 132: Fig. S1i is not informative since the cells are just too small to be seen.
- line 134: For the cobblestone morphology, the authors refer to Fig. 1f, which shows a gel.
- line 134: "in the absence of TGF- β ": should it not be "in the presence of TGF- β "?
- line 134: What does Fig. 1g show. TGF- β treatment or not? For how long?
- lines 139-143: The conclusion that correlates EPR expression with EMT-related factors in normal breast (Fig. S1m) makes me wonder if EMT occurs in normal breast tissue. Probably not? Not sure. Maybe we are looking at a mix of epithelial and stromal cells, rather than EMT.
- lines 166-167: The blot in Fig S2c has been cut off at the top in such a way that the EPRp in stomach cannot be seen, plus the cutoff is too close to the EPRp band anyway. Show more of the blot.
- lines 168-169: What does ACTB stand for in Fig. 2d?
- Related to Fig. 2g: Although not essential for the manuscript, the authors might consider showing what happens with the EPRp localization when cells are treated with TGF- β . It would be a good thing to show.
- line 242: "expectable" should be "expected".
- lines 252-255: the authors invite the reader to compare Fig. S3g with Fig. 3b; however, these are not comparable panels.
- lines 262 and 265: "either EPR or EPRSTOPE overexpression" should be changed to "overexpression of either EPR or EPRSTOPE". It is unclear the way it currently reads
- lines 303-314, related to Fig. 5b and Fig. S4c, d: The high level of Smad3 binding to the Cdkn1a promoter most likely reflects autocrine TGF- β signaling, which is then also probably the reason why the Smad3 association at the promoter seems constitutive. Blocking autocrine TGF- β signaling using SB431542 will most likely show the absence of Smad3 in the nucleus and at the promoter, and should reveal that the interaction of Smad3, and possibly of EPR is TGF- β -induced (in contrast to what we see now).
- lines 417-419: Please do not talk about the "TGF- β paradox" and the "switch". These terms are coined by those who do not have sufficient insight into the cancer biology of TGF- β , yet want to coin a quotable term. There is no paradox or switch.

- Throughout: "quantitate" should be "quantify".

We are grateful to the Reviewer for her/his efforts to substantially improve our manuscript and we have diligently worked for about two months in order to provide her/him with detailed answers as listed here below. The inclusion of new experimental data justifies the addition of two new Figures besides the new model Figure (new Figure 7) requested by Reviewer # 1.

General comments.

One question that still stands out, and that I steered towards in my previous comments, relates to the TGF- β -inducibility of this system. Based on what I see in the data and my experience, the EPR activity will be TGF- β dependent; however, the authors do not show convincing data in that regard. Indeed, the data without adding TGF- β seem to reflect a high level of autocrine TGF- β signaling. In other words, they often compare low with higher TGF- β signaling, rather than no TGF- β versus +TGF- β . Blocking (autocrine) TGF- β signaling is easily achieved using the T β RI kinase inhibitor SB431542, but this was not done. Comparing +SB431542 and +TGF- β would give much better results.

We followed the Reviewer's suggestion and performed a series of experiments using the inhibitor SB431542 to unambiguously demonstrate the ability of TGF- β to modulated EPR expression (please see our point-by-point answers).

Finally, in their rebuttal, the authors show me some data that would benefit the manuscript. Incorporating these into the manuscript would strengthen it. After all, I am reviewing it from the standpoint of a critical reader, and the reader does not consult the reviews.

According to the Reviewer's indication we have incorporated in this revised version some data that have been originally presented only to Reviewers (please see our point-by-point answers).

Point-by-point answers.

- line 103: Fig. S1b and its legend are insufficiently informative. How can I see that TGF- β modulates the interaction of KHSRP with 67 lncRNAs?

The reviewer is right. In order to make clear to readers the results of our transcriptome-wide studies we added two new Tables (Supplementary Table 1a and Supplementary Table 1b) that include the complete list of lncRNAs whose expression levels are influenced by TGF- β treatment (new Supplementary Table 1a) and a complete list of lncRNAs that interact with

KHSRP in a TGF- β -regulated manner (new Supplementary Table 1b). Consequently, we have deleted previous Supplementary Figures 1a and 1b.

- line 108: In Fig. S1c, what does P.I. stand for?

We are sorry for the inappropriate Figure legend. P.I. stands for Pre-Immune serum as now indicated in the Legend to the new Supplementary Figure 1a.

- line 108: In Fig. S1d, there is no control for GST-KHSRP. GST by itself (not shown) is normally not a good control for such experiments. GST fused to another protein should serve as a negative control.

We added a new panel (new Supplementary Figure 1b, left panel) showing that EPR does not interact with a negative control protein fused with GST.

- line 110: How do I know from looking at Fig. S1b that there is reduced interaction of EPR with KHSRP at 6 h?

We thank the Reviewer for allowing us to better show this point. The new Supplementary Table 1b clearly indicates that the interaction between KHSRP and EPR is significantly reduced upon 6 hours of TGF- β treatment. Further, Figure 5f show RIP-qPCR experiments yielding the same results.

- lines 111-113: The ChIP-qPCR in Fig. S1e is not convincing. Any controls? Also, the untreated sample shows very high Smad3 binding. Is this basal autocrine TGF- β signaling? If so, why does it decrease so much at 6 h? What is the level if you block TGF- β signaling using SB431542. These data do not look good, and do not allow for conclusions.

We followed the Reviewer's recommendation and performed experiments using the specific inhibitor of TGF- β receptor I signaling SB431542. As presented in the new Supplementary Figure 1c, SB431542 pre-treatment of NMuMG cells abrogated EPR down-regulation induced by TGF- β . In the same Figure, *Snai1* expression regulation by SB431542 is presented as positive control. These experiments also show that SB431542 treatment of cells not exposed to TGF- β (Time 0 in the left panel of the new Supplementary Fig. 1c) causes a limited increase of EPR levels that is not statistically significant. Thus, thanks to the Reviewer's question, we could rule out that basal autocrine TGF- β signaling plays a significant role in the modulation of EPR expression in NMuMG cells. Further, as requested

by the Reviewer, we present a positive control (*Serpine1*, lower section of panel d in the new Supplementary Fig. 1) and a negative control (*Mettl9* panel e in the same Figure) for the ChIP-qPCR experiments. Importantly —prompted by the Reviewer's criticisms— we adopted an alternative ChIP protocol in order to reduce the background signal (see the revised Methods Section and the new reference 55). The new ChIP protocol allowed us to improve the specific signal-to-noise ratio in NMuMG cells. As shown in the new Supplementary Fig. 1d, SB431542 treatment completely abrogated the TGF- β -dependent induction of SMAD3 interaction with both EPR and *Serpine1* promoters. The strong interaction of SMAD3 with the EPR promoter after 6 hours of TGF- β treatment (upper section of Supplementary Fig. 1d) allows us to hypothesize that a transcriptional repressor complex including SMAD3 might associate with the promoter region to rapidly down-regulate its EPR transcription. Future studies will be needed to obtain further molecular details on the transcriptional regulation of EPR gene expression by TGF- β signaling.

- As requested before, is the gene encoding EPR a direct target of TGF-b/Smad signaling? Is it induced (and then downregulated) in response to TGF-b + cycloheximide to block new protein synthesis? This should be a simple experiment.

To elucidate whether *de novo* protein synthesis is required for TGF- β -induced down-regulation of EPR expression, we performed experiments using cycloheximide. Data presented in the new Supplementary Fig. 1f indicate that the suppression of EPR expression by TGF- β does not involve *de novo* protein synthesis suggesting that, differently from *Zeb2/SIP1* (control in the new Supplementary Fig. 1f), EPR represents a direct target of TGF- β /SMAD signaling.

- line 117: Why do the authors refer to Fig. S1g? What is shown? I cannot figure it out from the legend, nor do I understand it.

We are sorry for not being clear enough. In the new Supplementary Figure 1h we show that (i) EPR is spliced as proved by the amplification of a qPCR product obtained using a couple of primers annealing with two distinct adjacent exons (flanking a ~20 Kb intron) and (ii) EPR is polyadenylated as revealed by the very similar yield of qPCR amplification using as template the products of two distinct reverse transcription reactions performed using either oligo-dT (to retro-transcribe only the poly-A tail) or random hexamers (as performed in the majority of the rest of the experiments). This is now thoroughly described in Methods Section and the Figure legend.

- line 117: *Fig. 1c has no controls. What are the controls for the different compartments? If Neat1 and tRNA-Lys are seen as controls, then it looks to me that the fractionation was not good at all. The standard controls for subcellular fractionation are needed.*

Also in this case, we are sorry for our lack of clarity. The method that we adopted in order to analyze RNA and proteins upon cell fractionation (Gagnon, K.T. *et al.* Analysis of nuclear RNA interference in human cells by subcellular fractionation and Argonaute loading. *Nat. Protoc.* **9**, 2045-2060 (2014), reference 51 in our revised manuscript), has been recently successfully utilized (Notzold L., *et al.*, The long non-coding RNA *LINC00152* is essential for cell cycle progression through mitosis in HeLa cells, *Sci. Rep.* **7**, 2265, 2017; Yen Y.P., *et al.*, Dlk1-Dio3 locus-derived lncRNAs perpetuate postmitotic motor neuron cell fate and subtype identity. *Elife.* **12**;7, 2018). We updated our list of controls with a predominantly nucleoplasmic RNA, *Rnu1a1*, and a predominantly cytoplasmic control, *Gapdh* mRNA (please see the new Figure 1c). Further, in order to strengthen our observation, we added an Immunoblot analysis utilizing protein extracts prepared in parallel to RNA samples and standard protein markers for cellular fractionation (HDAC1 as a chromatin-enriched protein, Tubulin Alpha [TUBA] as a cytoplasm-enriched protein, and hnRNPA1 as a nucleoplasm-enriched protein, please see the new Supplementary Figure 2a).

- lines 123-124: *Any possibility to show that EPR is made in other epithelia? For example, colon, which is high in Fig. 1b. The luminal epithelial expression is interesting, but that result has only a very narrow scope. It would be nice to see more evidence that it is expressed in epithelial or differentiated epithelial cells.*

Undoubtedly, this is an interesting point and our laboratory has another ongoing collaborative project on this topic. Thus, we deeply apologize for not being able to show details in the present manuscript. However, we would like to show here below to the Reviewer some preliminary results obtained by interrogating publicly available datasets of the Single-cell Mouse Cell Atlas (bis.zju.edu.cn/MCA/index.html). EPR is enriched in cells belonging to the columnar epithelium and the epithelium of villi in the small intestine, although in this tissue its expression is not exclusive of epithelial cells.

Small Intestine Cell Population	Gene	Avg. Difference	p Value
Epithelium of small intestinal villi (Fabp1_high)	EPR (BC030870)	0.6272	6.4908 e-59
Epithelium of small intestinal villi (Fabp8_high)	EPR (BC030870)	0.4748	1.0424 e-21
Columnar Epithelium (Small intestine)	EPR (BC030870)	0.4882	1.1745 e-11

Single cell mRNA sequencing was performed in mouse small intestine. Avg. Difference is the log2 fold expression change between the specified group and the other groups.

To this respect, we would like to thank again the Reviewer for pushing us to moderate our previous statement (epithelial-restricted) and to refer to EPR as an epithelial-enriched lncRNA.

- line 132: *Fig. S1i is not informative since the cells are just too small to be seen.*

We resized the panel (new Supplementary Figure 2c) in order to improve its visibility.

- line 134: *For the cobblestone morphology, the authors refer to Fig. 1f, which shows a gel.*

The Reviewer is right, the sentence was too compact and, as a consequence, misleading. We have re-written the sentence in order to avoid confusion.

- line 134: *“in the absence of TGF- β ”: should it not be “in the presence of TGF- β ”?* and

- line 134: *What does Fig. 1g show. TGF- β treatment or not? For how long?*

We apologize for the confusion. The Reviewer's remarks pushed us to completely re-write the sentence in order to clarify our points. Cells presented in Figure 1g were not treated with TGF- β .

- lines 139-143: *The conclusion that correlates EPR expression with EMT-related factors in normal breast (Fig. S1m) makes me wonder if EMT occurs in normal breast tissue. Probably not? Not sure. Maybe we are looking at a mix of epithelial and stromal cells, rather than EMT.*

We would like to better clarify our point. We wanted to remark that a statistically significant positive correlation exists between human EPR, CDH1, and OCLN expression while a negative correlation exists between human EPR, Vimentin, and SNAI1. The Results section

has been corrected in order to make clearer this point. As for the Reviewer's comment regarding the putative existence of either EMT or of a mixed epithelial/stromal population in normal breast tissue, it is difficult for us to reach a final conclusion. However, intrigued by the Reviewer's comment, we evaluated also datasets derived from Single cell RNA-Seq analysis performed in mice (<https://marionilab.cruk.cam.ac.uk/mammaryGland/>) and we *i*) confirmed that EPR is predominantly expressed in luminal cells and almost absent in basal and myo-epithelial cells of virgin mammary glands; *ii*) found that EPR expression is mutually exclusive with the expression of the EMT factor *Cdh2*. Single cell RNA-Seq analysis should rule out any stromal contamination of the epithelial cell population and the expression of *Cdh2* leaves open the possibility that cells with an intermediate epithelial/mesenchymal phenotype exist also in normal mammary gland.

- lines 166-167: The blot in Fig S2c has been cut off at the top in such a way that the EPRp in stomach cannot be seen, plus the cutoff is too close to the EPRp band anyway. Show more of the blot.

We replaced the Immunoblot presented in the previous Supplementary Figure 2c with a new one (please see the new Supplementary Fig. 3c) according to Reviewer's request.

- lines 168-169: What does ACTB stand for in Fig. 2d?

ACTB is Actin Beta in the HGNC guidelines nomenclature (<https://www.genenames.org>). We have now indicated the extended name in the Legends to all the Figures where it is shown.

- Related to Fig. 2g: Although not essential for the manuscript, the authors might consider showing what happens with the EPRp localization when cells are treated with TGF- β . It would be a good thing to show.

This is an interesting point worth of future investigations. We do not currently present data on EPRp localization upon TGF- β treatment for two reasons: *i*) the evaluation of EPRp localization in cells treated with TGF- β is difficult because the exposure to the cytokine rapidly down-regulates EPRp expression as demonstrated by Immunoblot analysis (please see Fig. 2d); *ii*) while the currently available anti-EPRp antibody works properly in Immunoblot experiments, it yields high background in Immunofluorescence. For this reason, we utilized anti-FLAG to detect FLAG-tagged EPRp in the experiments presented in Figure 2g and this fact prevents us from investigating Reviewer's question.

- line 242: “expectable” should be “expected”.

We have amended the mistake.

- lines 252-255: the authors invite the reader to compare Fig. S3g with Fig. 3b; however, these are not comparable panels.

The Reviewer is right. The sentence was not correct and we removed it. The purpose of the experiments presented in Supplementary Fig. 4g is to show that the expression changes induced in G1-enriched cells by overexpression of either EPR or EPRSTOPE are superimposable to those observed in the total cell population and, in order to make our point clearer to the readers, we ameliorated the new Supplementary Fig. 4g by showing the expression analysis of *Cdh1*, *Tjp1*, *Ocln*, and *Tnc* transcripts that are all included in Fig. 3b.

- lines 262 and 265: “either EPR or EPRSTOPE overexpression” should be changed to “overexpression of either EPR or EPRSTOPE”. It is unclear the way it currently reads

We have amended our mistake.

- lines 303-314, related to Fig. 5b and Fig. S4c, d: The high level of Smad3 binding to the *Cdkn1a* promoter most likely reflects autocrine TGF- β signaling, which is then also probably the reason why the Smad3 association at the promoter seems constitutive. Blocking autocrine TGF- β signaling using SB431542 will most likely show the absence of Smad3 in the nucleus and at the promoter, and should reveal that the interaction of Smad3, and possibly of EPR is TGF- β -induced (in contrast to what we see now).

We thank again the Reviewer for suggesting us to explore the possibility of an autocrine TGF- β signaling. First, also in this case, the improvement and refinement of our ChIP protocol allowed us to increase the specific signal-to-noise ratio of the SMAD3/*Cdkn1a* promoter interaction. Second, the results of ChIP experiments performed in cells treated with SB431542 demonstrate that the TGF- β -dependent enhancement of SMAD3 interaction with *Cdkn1a* promoter is blunted by the compound (please compare the new Fig. 5b with the new Supplementary Fig. 5d). Notably, SB431542 does not affect the increased levels of SMAD3/*Cdkn1a* promoter interaction observed in cells overexpressing either EPR or EPRSTOPE when compared to mock cells (new Supplementary Fig. 5d). Our interpretation of the existence of some interaction between SMAD3 and *Cdkn1a* promoter in untreated EPR-overexpressing cells—which is paralleled by the increased expression of *Cdkn1a*—is

that the interaction between overexpressed EPR and the *Cdkn1a* promoter might favor the recruitment of SMAD3 from the limited number of molecules already present in the nucleus of cells not treated with TGF- β . The presence of a limited but sizeable amount of SMAD molecules in the absence of any treatment is supported by some literature (Ref. 34). To be clearer, we completely re-wrote this part (last paragraph of page 11).

- lines 417-419: Please do not talk about the “TGF-b paradox” and the “switch”. These terms are coined by those who do not have sufficient insight into the cancer biology of TGF-b, yet want to coin a quotable term. There is no paradox or switch.

We followed Reviewer’s suggestions and removed the words “TGF- β paradox” and “switch”. Accordingly, we deleted previous reference 39 and substituted it with two new references (new Refs. 42 and 43).

- Throughout: “quantitate” should be “quantify”.

We have amended our mistakes.

REVIEWERS' COMMENTS:

Reviewer #2 (Remarks to the Author):

With their responses to the additional comments, the authors have improved the manuscript. While the results and conclusions based on individual sets of results do represent interesting new knowledge, the conglomerate of results and conclusion leaves me with a feeling that the mechanism has not been fully "nailed down". But this is as good as it will get, I believe.

Some targeted remarks that require action:

1. The new sentence (yellow) on lines 135-140 have internal repetition, fail to refer Fig. 1e, and, when taken together, seem to refer to data that are partially redundant. Critically evaluate what is written there, make it clearer and avoid showing redundant data.
2. The new Fig. 7 does not help, and is not needed. The problem is that the readers will take the model at face value, and I do not believe that this can be done at this time.
3. The newly inserted sentence (yellow) on lines 428-430 should be removed. It is just not right as written, although I know that some statements along this line are being made. It is not that simple.

REVIEWERS' COMMENTS:

Reviewer #2 (Remarks to the Author):

Some targeted remarks that require action:

1. The new sentence (yellow) on lines 135-140 have internal repetition, fail to refer Fig. 1e, and, when taken together, seem to refer to data that are partially redundant. Critically evaluate what is written there, make it clearer and avoid showing redundant data.

The sentence has been re-written according to Reviewer's suggestions.

2. The new Fig. 7 does not help, and is not needed. The problem is that the readers will take the model at face value, and I do not believe that this can be done at this time.

We have removed the Figure 7.

3. The newly inserted sentence (yellow) on lines 428-430 should be removed. It is just not right as written, although I know that some statements along this line are being made. It is not that simple.

We have removed the sentence.